

# A 100-Year record of mineralogical variations in Northeastern Greenland ice-core dust: Insights from individual particle analysis

Naoko Nagatsuka[1, 2], Kumiko Goto-Azuma[2], Kana Nagashima[1], Koji Fujita[3], Yuki Komuro[2, 4], Motohiro Hirabayashi[2], Jun Ogata[2], Kaori Fukuda[2], Yoshimi Ogawa-Tsukagawa[2], Kyotaro Kitamura[2], Ayaka Yonekura[5], Fumio Nakazawa[2], Yukihiko Onuma[6], Naoyuki Kurita[7], Sune Olander Rasmussen[8], Giulia Sinnl[8], Trevor James Popp[8] and Dorthe Dahl-Jensen[8]

[1] Research Institute for Global Change, Japan Agency for Marine-Earth Science and Technology, Yokosuka, 237-0061, Japan
[2] National Institute of Polar Research, Tokyo 190-8518, Japan
[3] Graduate School of Environmental Studies, Nagoya University, Nagoya 464-8601, Japan
[4] Laboratory for Environmental Research at Mount Fuji, Tokyo 169-0072, Japan
[5] Marine Works Japan Ltd., Kanagawa 237-0063, Japan
[6] Earth Observation Research Center (EORC), Japan Aerospace Exploration Agency (JAXA), Tsukuba, 305-8505, Japan
[7] Institute for Space-Earth Environmental Research, Nagoya University, Nagoya 464-8601, Japan
[8] Physics of Ice Climate and Earth, Niels Bohr Institute, University of Copenhagen, Tagensvej 16, DK-2200, Copenhagen N, Denmark

**Correspondence**: Naoko Nagatsuka (nagatsukan@jamstec.go.jp)

**Abstract**

Understanding the spatial and temporal variations in mineral dust sources in Greenland ice cores during the Holocene is challenging due to low dust concentration. Here, we present the first continuous records of the size and composition, as well as the temporal variations in potential sources, of mineral dust preserved in a northeastern Greenland ice core (EGRIP) covering the period from 1910 to 2013. Using a multi-proxy provenance approach based on individual particle analysis, we reconstruct variations in ice-core dust sources. We apply a recently developed provenance tracing technique, namely scanning electron microscopy (SEM)-cathodoluminescence (CL) analysis of single quartz particles, and SEM energy-dispersive X-ray spectroscopy (EDS) analysis to the Greenland ice core. The SEM-CL/EDS results reveal that the primary dust sources in the EGRIP ice core are Asian (Gobi Desert) and African (Sahara Desert) deserts. Their relative contributions have shifted since the 1970s–1980s: the contribution from the Gobi Desert has decreased whereas that from the Sahara Desert has increased. Our findings demonstrate that SEM-CL analysis is a valuable tool for identifying ice-core dust sources and reconstructing their variations during periods of low dust concentration. Additionally, we compare the EGRIP ice-core dust records with those of a northwestern Greenland ice core (SIGMA-D) to determine spatial variations in potential dust sources within the Greenland Ice Sheet over the past 100 years. The results reveal that the variability and mineral composition of the EGRIP ice-core dust differs significantly from those of the SIGMA-D ice-core dust, indicating that the dust in these two ice cores was likely transported from different geological sources. The SIGMA-D ice-core dust exhibits multidecadal variations, reflecting increased dust from the Greenland coastal region during warmer periods. Conversely, the EGRIP ice-core dust shows low temporal variation, suggesting a smaller contribution from local sources.

## 1. Introduction

The history of dust deposition on the Greenland Ice Sheet reconstructed from ice cores provides substantial insight into past climate and environmental changes. Ice-core mineral dust exhibits significant variations in concentration, particle size, and composition over different time scales. On glacial-interglacial timescales (e.g., from the Eemian to the Holocene), ice-core dust records show a strong correlation with climate variability, as indicated by $\delta^{18}O$ records. Dust concentrations in central Greenland ice cores increased by a factor of 100 during the Last Glacial Maximum compared to those in the Holocene and





show a strong correlation with temperature (Steffensen, 1997; Fuhrer et al., 1999; Schüpbach et al., 2018). On seasonal timescales, dust from a northwestern Greenland firn ice core (NEEM, Gfeller et al., 2014) as well as that from snow on the north (Summit, Whitlow et al., 1992; Drab et al., 2002), northeast (EGRIP, Nakazawa et al., 2021; Komuro et al., 2024),

central (NEEM, Kuramoto et al., 2011), and south Greenland Ice Sheet (SE-Dome, Oyabu et al., 2016) show seasonality characterized by a peak concentration in springtime during the Holocene. For the North Greenland Ice Core Project (NGRIP) ice core, the present-day dust concentration varies from >140 mg kg$^{-1}$ in the spring to >20 mg kg$^{-1}$ in the autumn (Bory et al., 2003a). Glacial-interglacial and seasonal variability may be related to the effects of climate change, environmental changes in dust sources such as their extent and aridity, and changes in atmospheric transport (Svensson et al., 2000).

Greenland ice-core dust can also be used to estimate variations in the mass balance of the ice sheet. Recently, areas of "dark ice" have appeared and expanded on the ice-sheet surface (Shimada et al., 2016). Such ice is thought to be partly responsible for the recent reduction in the albedo of the Greenland Ice Sheet and to have contributed to a recent increase in the melting rate (e.g., Wientjes et al., 2011; Alexander et al., 2014; Shimada et al., 2016). One possible reason for the expansion of dark-ice regions is an increase in the content of cryoconite, a mixture and/or aggregate of mineral dust and organic matter

produced by glacial microbes (e.g., Takeuchi et al., 2014; Chandler et al., 2015). Mineral dust deposited on the glacial surface is likely to be related to increased cryoconite content because such dust is the principal constituent of cryoconite and can affect microbial biomass, another component of cryoconite, by supplying nutrients. Nagatsuka et al. (2016) reported that englacial dust appears to be important for the formation of cryoconite on the dark ice surface of a glacier in northwestern Greenland. The englacial dust was deposited from the atmosphere in the past in the upstream, travelled through the ice sheet,

outcropped again in the ablation zone, and formed cryoconite. We assume that the dust was likely deposited widely on the ice sheet in the past and thus preserved in ice cores. Thus, reconstructing variations in the composition and sources of ice-core minerals that are expected to be exposed on the ice surface in the future is essential for evaluating the impact of dust on upcoming darkening events as well as on the ice mass loss of the Greenland ice sheet.

In Greenland ice core studies, potential dust sources have predominantly been identified through geochemical analyses,
particularly those based on Sr, Nd, Pb, and Hf isotope ratios. The isotope ratios for mineral dust in central Greenland ice cores, obtained by the Greenland Ice Sheet Project Two (GISP2) and the Greenland Ice Core Project (GRIP), suggest that the most probable sources of this dust are East Asian deserts and/or central European loess (Biscaye et al., 1997; Svensson et al., 2000; Újvári et al., 2015). North Africa is also considered as a potential source (Újvári et al., 2022). On the other hand, the isotope ratios for minerals found in an ice core obtained from southeastern coastal Greenland indicate that the dust originated

from local sources (Simonsen et al., 2019). However, isotope analyses require at least 1–3 mg of dust (e.g., Svensson et al., 2000; Újvári et al., 2022) and primarily target ice-core dust originating from glacial periods characterized by high dust concentration. For the NGRIP ice core, Ruth et al. (2003) found the highest dust concentrations (4–8 mg dust/kg ice) in stadial periods and relatively low concentrations (0.3–0.7 mg dust/kg ice) in interstadial periods. Several studies have attempted the measurement of the isotope ratios of ice-core dust from the Holocene (e.g., 8.2–11.6 ka: Han et al., 2018, 4–7

ka: Simonsen et al., 2019, 1997–2001 CE: Bory et al., 2003a), a period characterized by much lower dust concentration (less than 0.1 mg dust/kg ice in inland Greenland ice cores; Vallelonga and Svensson, 2014) than that in a glacial period. A single sample thus requires the accumulation of ice over tens to hundreds of years. Therefore, there is little high-temporal-resolution data about the sources and variability of mineral dust in the last several hundred years, during which global warming has progressed, and the environment has changed. To tackle this problem, Nagatsuka et al. (2021) analysed the

morphology and mineralogy of individual dust particles in a northwestern Greenland ice core using scanning electron microscopy (SEM) and energy-dispersive X-ray spectroscopy (EDS). Their study produced historical records of mineral dust sources for the past 100 years with a resolution of only 5 years, thus demonstrating the effectiveness of this approach for determining variations in the sources of ice-core mineral dust during periods of low dust concentration. However, unlike Sr,



Nd, Pb, and Hf isotope ratios, this method allows only estimates, not definitive identification, of the precise source regions of ice-core dust.

Ice-core dust sources probably vary geographically even within the Greenland Ice Sheet because environmental conditions, such as altitude and distance from the coast, differ greatly among regions (e.g., Bory et al., 2003b; Kjær et al., 2022). The interior of the ice sheet generally has the highest elevation (Bamber et al., 2013; Helm et al., 2014). Bory et al. (2003b) analysed samples from the 11th and 17th–18th centuries and reported that the dust transported over long distances was deposited at almost all elevated interior sites on the ice sheet, whereas it was likely that the dust from local ground sources was present only at coastal sites. This distinction seems to depend on the distance from the ice sheet margin and/or altitude (Bory et al., 2003b). Furthermore, atmospheric transport may also affect the geographical variations in mineral dust sources over the ice sheet. Based on the compositional variations of ice-core minerals and a back-trajectory analysis, it has been reported that the air mass at the northwestern ice sheet (SIGMA-D site) originated from the west coast of Greenland and northern Canada during the period 1958–2014 (Nagatsuka et al., 2021). On the other hand, the air mass and dust at the central ice sheet (GISP2, GRIP, and NGRIP sites) likely originated mostly from Asia, Europe, and North Africa during the period 45–12 ka BP as well as the 17th century (Svensson et al., 2000; Bory et al., 2003b; Újvári et al., 2015, 2022). The primary source of dust in southeastern Greenland (Dye-3 site) was estimated to be Asia, with North Africa as an additional source in the 18th century (Lupker et al., 2010). Although differences in geographical conditions may have led to nonuniform dust deposition over the ice sheet, spatial variations in the continuous records for the Greenland ice-core dust sources have still not been clarified. Since the SIGMA-D site is located relatively close to the margin of the ice sheet, local dust contributions are significant, making it difficult to identify variations in dust from distant sources (Nagatsuka et al., 2021). Therefore, identifying dust sources and their variations in ice cores from more inland sites with high temporal resolution should provide new insights into long-range dust transport to Greenland and its variations.

This paper describes the temporal and geographical variations in ice-core dust sources from the interior of the Greenland Ice Sheet. We present the first continuous records of the size and composition, as well as the temporal variations in potential sources, of mineral dust preserved in an ice core obtained from the northeastern Greenland Ice Sheet covering nearly 100 years (1910–2013). The East Greenland Ice-Core Project (EGRIP), which is an international ice coring project spearheaded by the University of Copenhagen, Denmark, was launched in 2015 as the first deep ice coring project in the northeastern inland ice sheet (Goto-Azuma et al., 2021). In this study, the size and mineral composition of dust particles in a shallow ice core from the EGRIP site are analysed using SEM and EDS. Additionally, we apply a recently developed provenance tracing method, namely SEM-cathodoluminescence (CL) analysis of individual quartz particles (Nagashima et al., 2017, 2023, in preparation), to the EGRIP ice core to improve the precision of ice-core dust source identification during periods of low dust concentration. SEM-CL detects native imperfections and impurities in quartz that vary with conditions during its formation and subsequent geological background (Zinkernagel, 1978), enabling the characterization of dust sources through the identification of bedrock types (volcanic, plutonic, and metamorphic rocks) with only a small sample amount. The results are compared to data from the northwestern coastal Greenland ice core (SIGMA-D), as analysed by Nagatsuka et al. (2021), to identify regional variations in dust sources within the ice sheet.

## 2. Samples and analytical methods

### 2.1 EGRIP ice core

The ice core was drilled at an elevation of 2708 m above sea level on the northeast side of the EGRIP deep drilling site (75.38° N, 36.00° W) in July 2017 (Fig. 1). The EGRIP site is located 470 km west of the eastern coast of Greenland and close to the onset of the Northeast Greenland Ice Stream, which is the largest ice stream on the Greenland ice sheet (Joughin et al., 2010; Vallelonga et al., 2014). The ice core was recovered from a depth of 1.51–133.09 m.






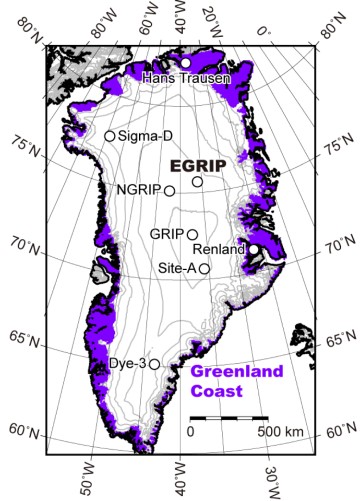

**Figure 1.** Location map of EGRIP ice core site (75.38° N, 36.00° W, 2708 m above sea level) and SIGMA-D, Hans Tausen, NGRIP, GRIP, Renland, Site-A, and Dye-3 ice core sites used for back-trajectory comparison. Contour lines are drawn at 500-m intervals. The purple shaded region denotes the ice-free coastal terrain (GrC).

## 2.2 Analyses of major elements, microparticles, and tritium in EGRIP ice core

To date the EGRIP ice core, the sodium (Na), sulphate ($SO_4$), and tritium concentrations were measured below a depth of 1.51 m. The Na and S concentrations were determined continuously along the core using an inductively-coupled plasma mass spectrometry (ICP-MS; 7700, Agilent Technologies, USA) system connected to a continuous-flow analysis (CFA)

system at the National Institute of Polar Research (NIPR). $SO_4$ concentrations were calculated from S concentrations under the assumption that the dominant S-containing species is $SO_4$. Details of the CFA analysis method are described in Goto-Azuma et al. (2024) and Goto-Azuma et al. (2025).

For tritium measurements, ice-core samples were taken at 0.04-m intervals at depths of 13–15 m. The tritium content was measured using a liquid scintillation counting technique at Nagoya University. The ice samples were distilled and mixed

with a liquid scintillation cocktail (Ultima Gold LLT, PerkinElmer, USA). Radioactivity associated with tritium was measured using a low-background liquid scintillation counter (Quantulus 1220, PerkinElmer, USA) for 1500 min. The lower detection limit was about 7 TU and the relative uncertainty was less than 5%.

For comparison with the mineral composition determined using SEM-EDS, the concentrations of elements that originated from terrestrial dust were measured using the same ICP-MS system as that used for Na and $SO_4$ measurement (Goto-Azuma

et al., 2024, 2025). These elements included silicon (Si), aluminium (Al), magnesium (Mg), potassium (K), calcium (Ca), and iron (Fe). Si, Al, and Fe predominantly originated from terrestrial dust, and some of the Mg, K, and Ca originated from sea salt. Under the assumption that Na solely originated from sea salt, we calculated the non-sea-salt (nss) fractions, which mainly originated from terrestrial dust, as follows:

$$[nssCa] = [Ca] - (Ca/Na)_{sea} \times [Na], \tag{1}$$

where $(Ca/Na)_{sea}$ is the mass ratio of Ca to Na in seawater, which is 0.037 (Savarino and Legrand, 1998). Similarly, the concentrations of nssMg and nssK were calculated using the mass ratio of Mg to Na (0.12) and that of K to Na (0.037) in





seawater, respectively (Savarino and Legrand, 1998). On average, the nss fractions of Ca, Mg, and K accounted for 96%, 67%, and 88% of the total concentrations of these elements, respectively.

For dust-particle size and concentration measurements, a portion of the meltwater was collected from the CFA samples taken at depth intervals of 0.12 m using a fraction collector. The concentration of dust was measured by a Coulter counter (Multisizer 4e, Beckman Coulter, USA) using a 30-μm-aperture tube in a Class 10000 clean room at the NIPR. The size bins covered the range of 0.7 to 18.0 μm.

## 2.3 SEM-EDS analysis of mineral dust

To extract mineral dust from the ice core for SEM analysis, we first melted the ice-core samples and collected them in pre-cleaned glass vials in 10-year intervals. Next, a 500-μL aliquot from each well-mixed sample was filtered through a polycarbonate membrane filter (Advantec) with a diameter of 25 mm and a pore size of 0.1 μm in a Class 10000 clean room. We determined the size and chemical composition of individual mineral dust particles using SEM (Quanta FEG 450, FEI) and EDS (X-Max 50, Oxford Instruments, UK) at the NIPR. The filter was attached to aluminium stubs using carbon tape and coated with a layer of platinum. Under the assumption of a uniform distribution of dust samples on the filter, we divided the filter into four equal sections and observed a total of 200 randomly selected particles from one of these sections. The equivalent circle diameter was measured using image-processing software (ImageJ, National Institutes of Health, USA). To minimize bias in the selection of mineral dust, particles were observed in various positions on one quarter of the filter over a span of a few days. Particle number size distributions and log-normal fitting results (fitting curves, mode diameters, and half peak width) were calculated using Igor Pro software (WaveMetrics, USA). The content levels of major elements and related oxides were determined from the EDS spectra. Details of the SEM-EDS analytical method are described in Nagatsuka et al. (2021).

## 2.4 Mineral identification

Mineralogical identification of the EGRIP ice-core dust was performed using three procedures (Nagatsuka et al., 2021): (1) matching of the spectral pattern for each particle to those for mineral standards (Severin, 2004); (2) comparison of the morphology and oxide composition of the ice-core dust to those of mineral standards; (3) application of the elemental peak intensity ratio sorting scheme used to identify Greenland ice-core (GISP2) mineral dust (Donarummo et al., 2003). The results of previous studies have demonstrated that a comparison of the results obtained using these procedures enables reliable identification of ice-core minerals (Maggi et al., 1997; Wu et al., 2016).

Most of the silicate minerals in the EGRIP ice core were categorized into five types (Types A–E) based on the formation environment, formation process, and possible sources, as previously done for the SIGMA-D ice core (Nagatsuka et al., 2021). The details of each mineral type are given in Table 1. These minerals have localized distributions (Ito and Wagai, 2017). The relative abundance for pairs of mineral types reflects the relative contributions of chemical and physical weathering processes and is thus an indicator of latitude (e.g., Biscaye, 1965; Griffin et al., 1968; Biscaye et al., 1997; Maggi, 1997; Svensson et al., 2000; Donarummo et al., 2003). The variations in mineral composition among the five mineral types reflect the climatic and geological conditions in their source areas and can be therefore used as indicators for the dust source and the transportation process in different periods.



**Table 1.** Description of silicate mineral types in EGRIP ice core

| Type | Minerals | Notes | Possible sources | References |
|------|----------|-------|------------------|------------|
| Type A | Kaolinite and other kaolin minerals (nacrite, dickite, halloysite) and/or pyrophyllite | Clay minerals composed of Si and Al generally formed from the primary and secondary minerals in warm and humid regions by chemical weathering. | Low- to middle-latitude areas (e.g., Central Africa and Southeast Asia) and relict deposits of past warmer climates (e.g. Tertiary Northern Canadian deposits) | Mueller and Bocquier, 1986; Velde, 1995; Bergaya et al., 2006; Nagatsuka et al, 2021 |
| Type B | Micas, chlorite, and their mixture | Clay minerals formed in cold and dry regions by mechanical weathering. | High-latitude (e.g., North America, Russia, North Europe, Greenland) and/or desert areas (e.g., Asia and North Africa) | Cremaschi, 1987; Pye, 1987; Velde, 1995 |
| Type C | Feldspars (Na/Ca/K-plagioclase and K-feldspar) | Minerals formed in cold and dry regions by mechanical weathering. | High-latitude (e.g., North America, Russia, North Europe, Greenland) and/or desert areas (e.g., Asia and North Africa) | Nahon, 1991; Bory et al., 2003b |
| Type D | Mafic minerals | Minerals formed by mechanical weathering and less common in atmospheric dust. | Local areas (Greenland) | Deer et al., 1993 |
| Type E | Quartz | Most resistant to weathering processes at the Earth's surface and whose abundance in the atmosphere is likely related to desert source areas. | Desert areas (e.g., Asia and North Africa). | Pye, 1987; Yokoo et al., 1994; Genthon and Armengaud, 1995) |



### 2.5 SEM-CL analysis of mineral dust

To identify potential sources of the ice-core dust, we performed SEM-CL analysis on single quartz particles. First, we randomly selected 52-56 quartz particles on each filter for five different periods (1940–1950, 1970–1980, 1980–1990, 1990–2000, 2010–2013) from the EGRIP ice core using SEM (Quanta FEG 450, FEI) and EDS (EDAX, Octane Elite 30) at the Japan Agency for Marine-Earth Science and Technology (JAMSTEC). Then, we measured their CL spectrum using SEM combined with a CL system (Gatan, Mono CL4 Swift) at JAMSTEC. As one of the references for North America, a potential

source area of dust, we also analysed quartz particles (38 particles) in a surface dust sample (cryoconite) from a mountain glacier in Alaska (Gulkana Glacier, 63°14′N, 145°42′E, Fig A1). This sample was analysed for Sr-Nd isotope ratios and denoted as S5 in Nagatsuka et al. (2019). The isotopic values were similar to those of Alaskan loess, suggesting an Alaskan sedimentary origin. Details of the SEM-CL measurement method are described in Nagashima et al. (2023).

The measured CL spectrum was separated into six Gaussian emission components (EC1–EC6), centred at 1.75, 1.9, 2.0, 2.2,

2.7, and 2.9–3.2 eV, respectively (Table A1), to identify the type and intensity of emission components shown as Gaussian curves in energy units. These emission components are the same as those identified in Nagashima et al. (2023). We calculated the relative intensity of each component using the fractional area of each Gaussian curve and the fractional areas of EC1–EC6 for all measured CL spectra. Nagashima et al. (2023) performed a cluster analysis based on Ward's method using the calculated fractional areas of EC1–EC6 for quartz particles from Asian deserts and seawater filtration samples,

which represented a wide range of terrestrial bedrocks. Their calculated cluster analysis dendrogram contained three clusters (Clusters 1–3). Following Nagashima et al. (2023), we classified the CL spectrum of each quartz particle in EGRIP filters into Clusters 1–3 by calculating the sum of the squared differences between the EC1–EC6 fractional areas of the CL spectrum for each quartz particle and those of Clusters 1–3 and then selecting the cluster with the minimum difference.

### 2.6 Back-trajectory analysis

The air mass transport pathways to the EGRIP site were analysed using the Hybrid Single-Particle Lagrangian Integrated Trajectory (HYSPLIT) model (Stein et al., 2015). The initial air masses were set to 50, 500, 1000, and 1500 m above ground level at the EGRIP site for the 20-day back-trajectories. The probability distribution for the air mass within this altitude range (< 1500 m above ground level) was calculated with 1° × 1° degree resolution. We assumed that the dust particles had been deposited via a wet process (Iizuka et al., 2018; Parvin et al., 2019), for which the probability was weighted by the

daily precipitation on the date when the air mass arrived at the ice core site (Nagatsuka et al., 2021). We used daily precipitation from the ERA5 reanalysis dataset produced by the European Centre for Medium-Range Weather Forecasts (Hersbach et al., 2020). The regional contribution was also calculated from the probability distribution, for which land areas were divided into nine regions: the Greenland Ice Sheet (GrIS), the Greenland coast (GrC), North America (NA), Europe (EU), Russia (RUS), Central Asia (CA), Southeast Asia (SA), Middle East (ME), and Africa (AF) (Fig. 2a). Considering the

dust source, the ocean and GrIS were excluded from the calculation.



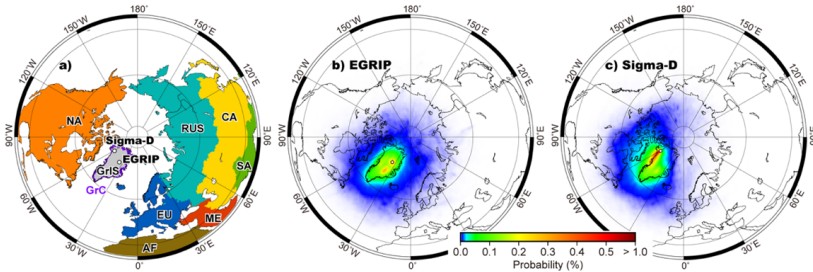

**Figure 2.** Map showing (a) location of EGRIP and SIGMA-D ice core sites in Greenland and nine regions used for calculating regional contribution (GrIS: Greenland Ice Sheet, grey; GrC: Greenland coast, purple; NA: North America, orange; EU: Europe, blue; RUS: Russia, light blue; CA: Central Asia, yellow; SA: Southeast Asia, green; ME: Middle East, red; and AF: Africa, brown) and probability distribution for air mass at (b) EGRIP and (c) SIGMA-D sites from 7-day three-dimensional back-trajectory analysis from 1958 to 2014. The probability distribution was calculated by summing the air masses at 50, 500, 1000, and 1500 m above ground level.

**2.7 Snow cover fraction**

To examine the surface conditions for adjacent source areas of mineral dust in the EGRIP ice core, we analysed the changes in the inter-annual snow cover fraction obtained from numerical simulations using climate models. Global climate models have been used by international organizations when performing numerical simulations to reproduce or predict past or future climate change. The results obtained in these studies were compiled and published by World Climate Research Programme Coupled Model Intercomparison Project Phase 6 (CMIP6; Eyring et al., 2016). To investigate the sources of the EGRIP ice-core dust, we used snow cover fractions derived from a land-historical experiment with climate models for the period 1850–2014 in the Land Surface, Snow and Soil Moisture Model Intercomparison Project (LS3MIP; van den Hurk et al., 2016) of CMIP6. The experiment referred to as land-hist in LS3MIP was conducted using various land surface models, which in climate models simulate physical processes at the land surface, with prescribed atmospheric conditions near the land surface using the reanalysis dataset GSWP3 (Kim, 2017). In this study, we analysed the snow cover fractions simulated with twelve climate models, namely BCC-CSM2-MR, CESM2, CMCC-ESM2, CNRM-CM6-1, CNRM-ESM2-1, GFDL-ESM4, GISS-E2-1G, HadGEM3-GC31-LL, IPSL-CM6A-LR, MIROC6, MPI-ESM1-2-LR, and UKESM1-0-LL (Malyshev et al., 2018; NASA, 2018; Boucher, 2019; Danabasoglu, 2019; Stacke et al., 2019; Voldoire, 2019a, 2019b; Zhang et al., 2019; Onuma and Kim, 2020; Peano et al., 2020; Wiltshire, 2020; Wiltshire et al., 2020). Data for the inter-annual variations in snow cover fraction during summer on the northeast and southeast coasts of Greenland (boundary at 70° N) were obtained in the same manner as in Nagatsuka et al. (2021). Data for the inter-annual variations in snow cover fraction during summer on the northeast and southeast coasts of Greenland (boundary at 70° N) were obtained in the same manner as in Nagatsuka et al. (2021), but the ensemble means of the twelve models were used for the present study.

**3. Results**

**3.1 Dating of EGRIP ice core**

Ice core dating was performed by counting annual layers of Na concentration, which showed clear seasonal variations (Fig. A2). The observed seasonality of the water stable isotope ratio and chemical components in the ice cores and snowpack has been previously reported at various sites on the Greenland Ice Sheet and is often used for annual layer counting (Whitlow et



al., 1992; Legrand and Mayewski, 1997; Kuramoto et al., 2011; Oyabu et al., 2016; Kurosaki et al., 2020; Komuro et al., 2021; Nakazawa et al., 2021; Nagatsuka et al., 2021; Sinnl et al., 2022). The winter season in the EGRIP ice core was defined as the depth at which the Na concentration had a maximum value. The period from winter season to winter season was counted as 1 year.

Three fixed dates were provided by the ice layers, tritium profile, and $SO_4$ spikes, respectively. The ice layer at a depth of
2.14 m is assumed to correspond to summer 2012 based on a snow pit observation at EGRIP in summer 2017 (Komuro et al., 2021). We interpreted the ice layer at a depth of 2.40 m as a layer that formed via the refreezing of meltwater produced by surface melting in summer 2012. The sharp tritium peak at a depth of 14.1 m corresponds to nuclear bomb testing in 1963 (Koide et al., 1982; Clausen and Hammer, 1988). The two sharp $SO_4$ peaks at depths of 23.18 and 23.30 m are presumed to be associated with the eruption of the Katmai volcano, Alaska, which occurred on June 6–8, 1912 (e.g., Hildreth, 1983;
Hildreth and Fierstein, 2000; Sinnl et al., 2022). We assigned the latter peak to this eruption because this peak appears after the Na concentration peak (winter) and corresponds to the timing of the eruption in early summer. The $SO_4$ peak at a depth of 23.18 m, which is assumed to correspond to the 1913 winter/spring, is interpreted as stratospheric fallout of sulphate aerosols produced by the same eruption, although it may also be derived from anthropogenic sulphate, peaking in winter to early spring (Beer et al., 1991, Kuramoto et al., 2011; Oyabu et al., 2016) and/or the eruption of the Hekla volcano,
Iceland, starting on April 24, 1913 (Bigler et al., 2002). Volcanic aerosols from the 1912 Katmai eruption, known as the world's largest 20th-century volcanic eruption, were injected into the stratosphere and remained suspended until at least as late as December 1914 (e.g., Volts, 1975a, 1975b; Hildreth and Fierstein, 2012; Burke et al., 2019). Volcanic sulphate deposits from the stratosphere have also been found in the 1913 layer of other Arctic ice cores and are identified by a subsidiary $SO_4^{2-}$ peak (Yalcin et al., 2007; Burke et al., 2019). From a comparison of the annual layer counts and these
reference horizons, we estimated that the ice core dating uncertainty is 1 year. We also compared our EGRIP chronology to the Greenland ice-core chronology 2021 (GICC21, Sinnl et al., 2022) based on multiple ice cores. The two chronologies are in good agreement within their respective uncertainties. To confirm the ages in the bottom section of the analysed ice-core sample, we also used the ice layers at depths of 27.32 and 27.33 m, which are assumed to correspond to summer 1889. These layers have been identified in multiple ice cores from central Greenland (Alley and Koci, 1988; Clausen et al., 1988; Meese
et al., 1994) and northwest Greenland (Fischer et al., 1998). They are known to have formed via widespread surface melting events on the Greenland Ice Sheet in 1889 (Keegan et al., 2014).

From these analyses, we estimated that a depth of 23.64 m corresponds to the year 1910 and that the average accumulation rate for the period 1910–2013 was 0.11 m w.e. yr$^{-1}$. This rate is close to that for the period 1607–2011 at EGRIP reported by Vallelonga et al. (2014) (0.10 m w.e. yr$^{-1}$). Based on our chronology, the dates associated with the $SO_4$ peaks at 2.56, 9.20,
and 23.17 m were estimated to be 2011, 1987, and 1913, respectively. The existence of volcanic signals in these years is consistent with the results of previous ice core studies at EGRIP (Kjær et al., 2016, 2022).

### 3.2 Particle morphology

Figure 3 shows SEM images of various types of dust particle observed in the EGRIP ice core. The number size distribution indicates that the diameter of most particles was smaller than 2 μm (Table 2, Figs. 4 and A3), similar to values reported for
other Greenland ice cores (e.g., Steffensen, 1997; Biscaye et al., 1997; Nagatsuka et al., 2021). The mean particle diameter, calculated as 10-year-averaged values, ranged from 0.85 to 1.36 μm and the maximum particle diameter ranged from 5.06 to 8.94 μm. The size distribution varied depending on the period. The samples from 1910 to 1990 (except for samples from 1930 to 1940 and 1970 to 1980) showed a slightly bimodal distribution with smaller peaks at 0.32 to 0.66 μm and larger peaks at 1.10 to 3.34 μm. On the other hand, the samples from 1990 to 2013 showed a unimodal distribution with peaks at
0.71 to 0.76 μm (Fig. A3). No samples contained particles larger than 10 μm.





**Table 2.** Description of EGRIP ice-core dust samples

| Period | Ice | | | | Dust (particle size) | | | |
| | Top (m) | Bottom (m) | Average (μm) | Maximum (μm) | | Log-normal mode (μm) | | |
| | | | | | | First mode | Second mode | |
|---|---|---|---|---|---|---|---|---|
| 1910−1920 | 21.95 | 23.64 | 1.36 | 8.25 | | 0.52 | 1.46 | |
| 1920−1930 | 20.28 | 21.95 | 0.96 | 8.55 | | 0.56 | 3.34 | |
| 1930−1940 | 18.25 | 20.28 | 0.89 | 5.59 | | 0.66 | NA | |
| 1940−1950 | 16.55 | 18.25 | 0.95 | 8.27 | | 0.40 | 1.16 | |
| 1950−1960 | 14.80 | 16.55 | 0.88 | 7.60 | | 0.32 | 1.10 | |
| 1960−1970 | 12.60 | 14.80 | 1.10 | 8.29 | | 0.65 | 1.93 | |
| 1970−1980 | 10.79 | 12.60 | 0.85 | 7.78 | | 0.57 | NA | |
| 1980−1990 | 8.54 | 10.79 | 1.06 | 5.06 | | 0.65 | 1.93 | |
| 1990−2000 | 5.84 | 8.54 | 1.06 | 6.94 | | 0.71 | NA | |
| 2000−2010 | 3.01 | 5.84 | 1.07 | 8.94 | | 0.76 | NA | |
| 2010−2013 | 1.93 | 3.01 | 0.99 | 8.72 | | 0.75 | NA | |

*NA: not available*

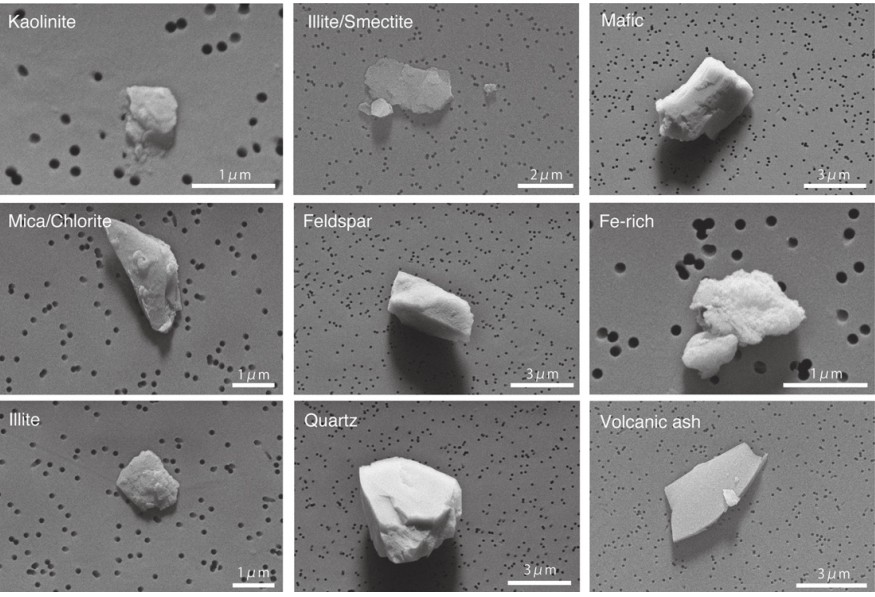

**Figure 3.** SEM micrographs of samples from each mineral group in EGRIP ice core.





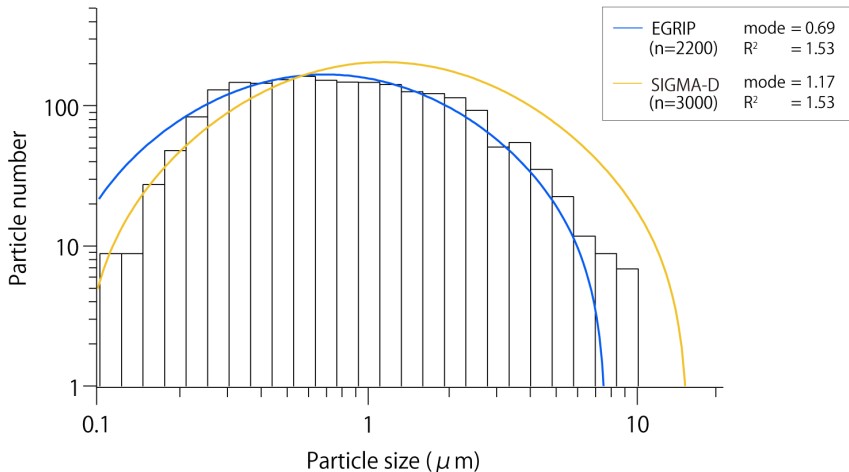

**Figure 4.** Size distribution and log-normal fitting results (mode: mode diameter; $R^2$: half peak width) for minerals in EGRIP ice core and SIGMA-D ice core (fitting results only) samples during periods of 1910 to 2013 and 1915 to 2013, respectively.

### 3.3 Quantitative estimation of mineral dust


The EDS analysis showed that for all samples, the ice-core dust was composed predominantly of silicate minerals (94–98%, Fig. 5). The silicates were categorized as quartz, mafic minerals, Na/Ca-, K-, and Na/K-feldspars, clays (kaolinite, other kaolin minerals (nacrite, dickite, halloysite) and/or pyrophyllite (hereafter referred to as high-Si+Al minerals), smectite, illite, micas, and chlorite, as well as mixed layers of illite/smectite and micas/chlorite) (Figs. 3 and 6). We also observed a

few minerals with an identical rhyolitic volcanic glass composition, which have been found in other Greenland ice cores (e.g., Cook et al., 2022).

Based on the semi-quantitative EDS analysis, the proportion of the mica/chlorite mix was the highest (21–37%) and that of smectite was the lowest (0–2%) for all silicate mineral particles in nearly every period (Table 3 and Fig. 6). The proportions of high-Si+Al minerals, illite/smectite mix, illite, and quartz were the second highest, varying in the ranges 9–22%, 4–14%,

3–16%, and 4–12%, respectively.

The silicate mineral composition varied among the samples. Lower high-Si+Al mineral content and higher illite/smectite mix and illite content were found for the 1910–1970 samples (high-Si+Al minerals: 9–14%, illite/smectite mix: 6–14%, illite: 9–16%) compared to those for the 1970–2013 samples (high-Si+Al minerals: 18–22%, illite/smectite mix: 4–5%, illite: 3–13%). A higher volcanic glass content (11%) was also found in the 1910–1920 sample compared to that in samples from

other periods (0–2%). The compositions of other silicate minerals were similar for the same periods.

The ice-core samples also contained non-silicate minerals, mainly an Fe-dominant mineral, identified as an Fe oxide (hematite, magnetite, or pyrite, Fig. 5), at low concentrations. The relative abundance of these minerals was 2–6% and showed low variation among the samples.




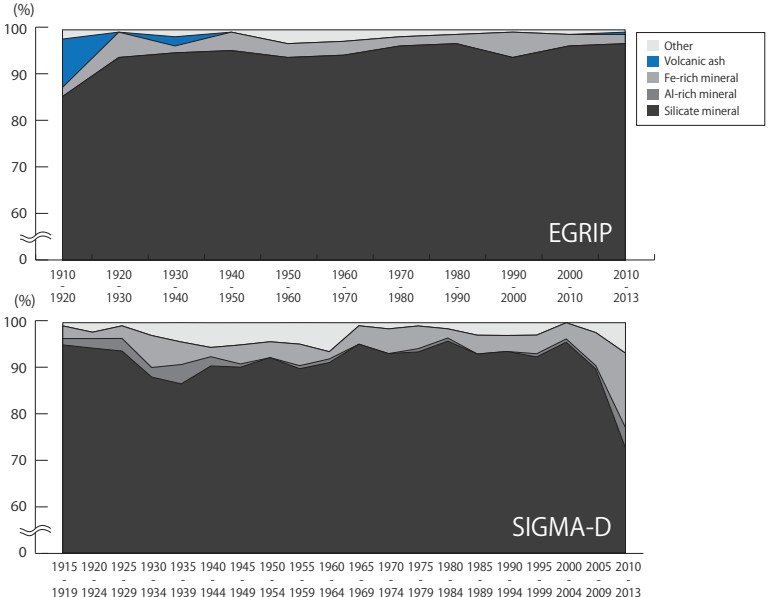


**Figure 5.** Variations of insoluble mineral records for EGRIP ice core (top) and SIGMA-D ice core (bottom) dust with 10- and 5-year resolutions, respectively.

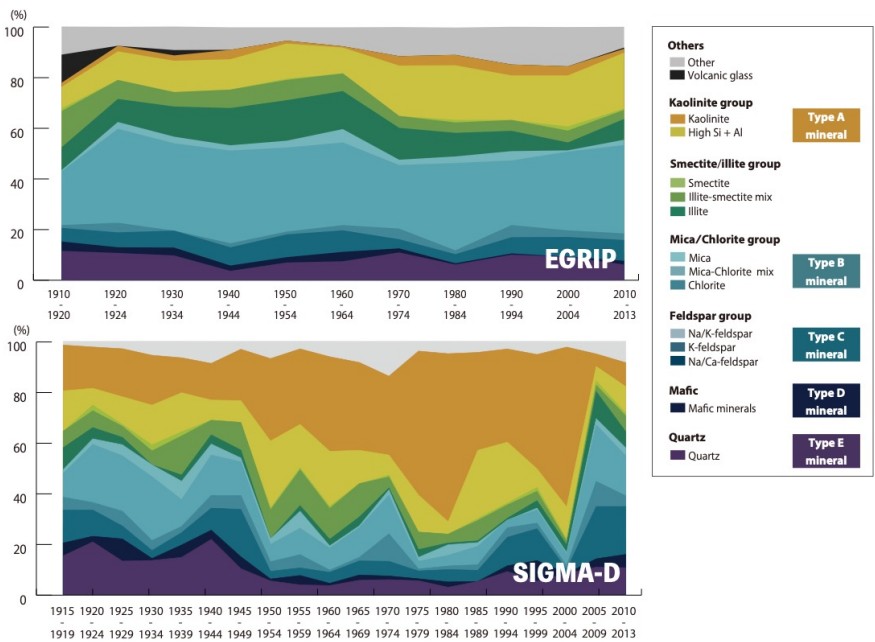

**Figure 6.** Variations of silicate mineral compositions in EGRIP ice core (top) and SIGMA-D ice core (bottom) dust with 10- and 5-year resolutions, respectively. "Feldspar group" includes Na/K-feldspar, K-feldspar, and Na/Ca-feldspar.



**Table 3.** Relative abundance (%) of silicate mineral groups for each EGRIP sample


| Sample period | Kaolinite | High-Si+Al | Smectite | Illite/Smectite | Illite | Micas | Micas/Chlorite | Chlorite | Na/Ca-feldspar | K-feldspar | Na/K-feldspar | Mafic | Quartz | Volcanic glass | Un-known |
|---|---|---|---|---|---|---|---|---|---|---|---|---|---|---|---|
| 1910–1920 | 1.6 | 8.5 | 1.1 | 14.3 | 9.0 | 0.5 | 21.2 | 1.1 | 3.2 | 1.1 | 1.1 | 3.7 | 11.6 | 11.1 | 11.1 |
| 1920–1930 | 2.2 | 11.3 | 0.0 | 7.5 | 9.1 | 2.7 | 37.1 | 3.8 | 4.3 | 1.6 | 0.0 | 2.2 | 10.8 | 0.0 | 7.5 |
| 1930–1940 | 2.1 | 12.6 | 0.0 | 5.7 | 11.9 | 2.6 | 34.5 | 0.0 | 5.2 | 1.0 | 0.5 | 3.1 | 9.8 | 2.1 | 9.3 |
| 1940–1950 | 3.7 | 12.0 | 0.0 | 7.3 | 14.7 | 2.1 | 36.6 | 1.6 | 3.1 | 3.1 | 1.0 | 2.1 | 3.7 | 0.0 | 8.9 |
| 1950–1960 | 1.1 | 13.9 | 0.5 | 8.0 | 16.0 | 2.7 | 33.2 | 1.1 | 8.0 | 0.5 | 0.5 | 2.1 | 7.0 | 0.0 | 5.3 |
| 1960–1970 | 0.5 | 10.2 | 0.0 | 7.0 | 15.0 | 5.3 | 32.6 | 2.1 | 4.3 | 2.1 | 2.1 | 3.7 | 7.5 | 0.0 | 7.5 |
| 1970–1980 | 3.7 | 19.9 | 0.0 | 4.7 | 12.6 | 2.1 | 25.1 | 4.2 | 2.6 | 0.0 | 1.0 | 1.6 | 11.0 | 0.0 | 11.5 |
| 1980–1990 | 4.1 | 21.6 | 1.0 | 4.1 | 9.3 | 2.6 | 34.5 | 1.5 | 2.6 | 1.0 | 0.0 | 0.5 | 6.2 | 0.0 | 10.8 |
| 1990–2000 | 4.3 | 17.6 | 0.0 | 4.3 | 8.0 | 3.7 | 25.5 | 4.8 | 4.8 | 1.6 | 0.0 | 0.5 | 10.1 | 0.0 | 14.9 |
| 2000–2010 | 3.6 | 20.2 | 1.6 | 4.7 | 3.1 | 0.5 | 31.1 | 2.6 | 6.2 | 0.0 | 1.6 | 0.0 | 9.3 | 0.0 | 15.5 |
| 2010–2013 | 1.5 | 22.1 | 0.5 | 3.6 | 8.2 | 2.1 | 34.9 | 2.6 | 5.1 | 2.6 | 0.5 | 1.5 | 6.2 | 0.5 | 8.2 |




### 3.4 Cluster compositions of cathodoluminescence spectra for quartz particles in EGRIP samples

The quartz particles in the EGRIP ice-core samples from 1940 to 1950 and 1980 to 1990 show a dominant contribution of Cluster 1 (57%) with lesser contributions of Cluster 2 (27–30%) and Cluster 3 (13–16%) (Fig. 7). On the other hand, the quartz particles in the samples from 1970 to 1980, 1990 to 2000, and 2010 to 2013 exhibit slightly lesser and higher

contributions of Cluster 1 (50–52%) and Cluster 2 (31–36%), respectively, and a similar contribution of Cluster 3 (13–19%) compared to those for the former two samples. Based on the empirical relations between the CL features of quartz particles and their associated host-rock types (Zinkernagel, 1978; Götze et al., 2001; Boggs et al., 2002; Götte and Richter, 2006), Nagashima et al. (2017, 2023) interpreted Clusters 1–3 as indicative of quartz particles in low-grade metamorphic or slow-cooling high-grade metamorphic rocks, plutonic rocks, and volcanic or rapidly cooled high-grade metamorphic rocks,

respectively. According to their interpretations, the first two samples are dominantly supplied from metamorphic rocks. This cluster composition is similar to that of an East Asian desert, especially the Gobi Desert. The other three samples show a rather larger contribution from plutonic rocks, which is similar to the cluster compositions of the Sahara Desert.

## 4. Discussion

### 4.1 Potential sources of EGRIP ice-core minerals

The EGRIP ice-core dust samples from 1910 to 2013 primarily consisted of silicate minerals, the most common mineral group in Earth's crust (Deer et al., 1993). The silicate mineral compositions in the EGRIP ice core have notably low variations among the samples (Figs. 6 and 8), indicating that the sources remained relatively stable during this period.

To identify the ice-core dust sources, we applied SEM-CL analysis for the first time in Greenland ice-core dust provenance research. SEM-CL has allowed the characterization of quartz particles from Asian and African deserts in seawater and a

mountain ice core (Nagashima et al., 2023, in preparation). The EGRIP ice-core quartz particles exhibit cluster compositions closely resembling those of the Gobi and Sahara Deserts (Fig. 7), suggesting that these Asian and African deserts are likely the primary sources of the ice-core dust. This finding aligns with findings for other inland Greenland ice-core dust during glacial periods (e.g., Biscaye et al., 1997; Svensson et al., 2000; Han et al., 2018; Újvári et al., 2022) based on Sr, Nd, and Hf isotope ratios. These results also confirm that SEM-CL cluster analysis is a valuable tool for identifying

ice-core dust sources during periods of low dust concentration.

The size and composition of the EGRIP ice-core mineral dust further supports the dust contribution from these two regions. The ice-core dust is mostly smaller than 2 μm (Figs. 4 and A3), suggesting that it predominantly consists of particles transported over long distances. A ternary clay mineralogy diagram, showing the proportions of illite/micas/chlorite, kaolinite, and smectite, reveals that EGRIP ice-core dust samples are similar to loess from Central and Southeast Asia, as

well as to dust from other Greenland ice cores (GRIP, Svensson et al., 2000; GISP2, Maggi, 1997). On the other hand, the EGRIP ice core shows a similar kaolinite proportion but slightly lower smectite proportion compared to those of the Sahara Desert (Fig. 9). This difference can be explained by smectite fractionation during atmospheric transport (Scheuvens et al., 2013; Singer et al., 2004). The Sahara thus remains a plausible dust source.



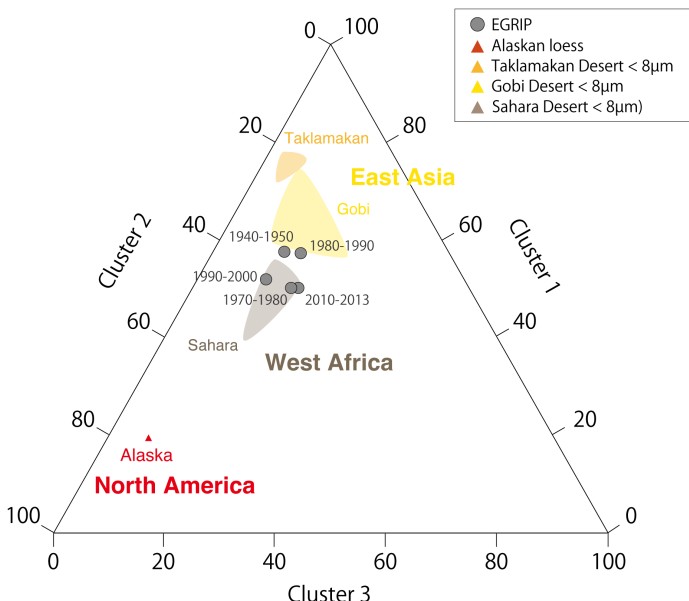


**Figure 7.** Cluster composition of quartz particles in the EGRIP ice core from five different periods (1940–1950, 1970–1980, 1980–1990, 1990–2000, 2010–2013) and those from Gobi, Taklamakan, and Sahara Deserts (Nagashima et al., in preparation) and North America (Alaska).

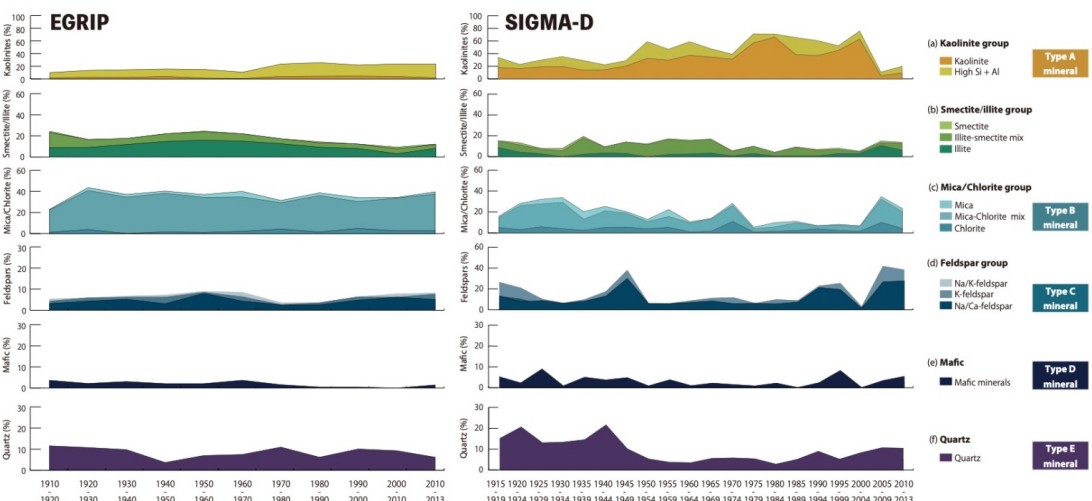

**Figure 8.** Comparison of historical records of silicate mineral proportions for EGRIP and SIGMA-D ice cores.



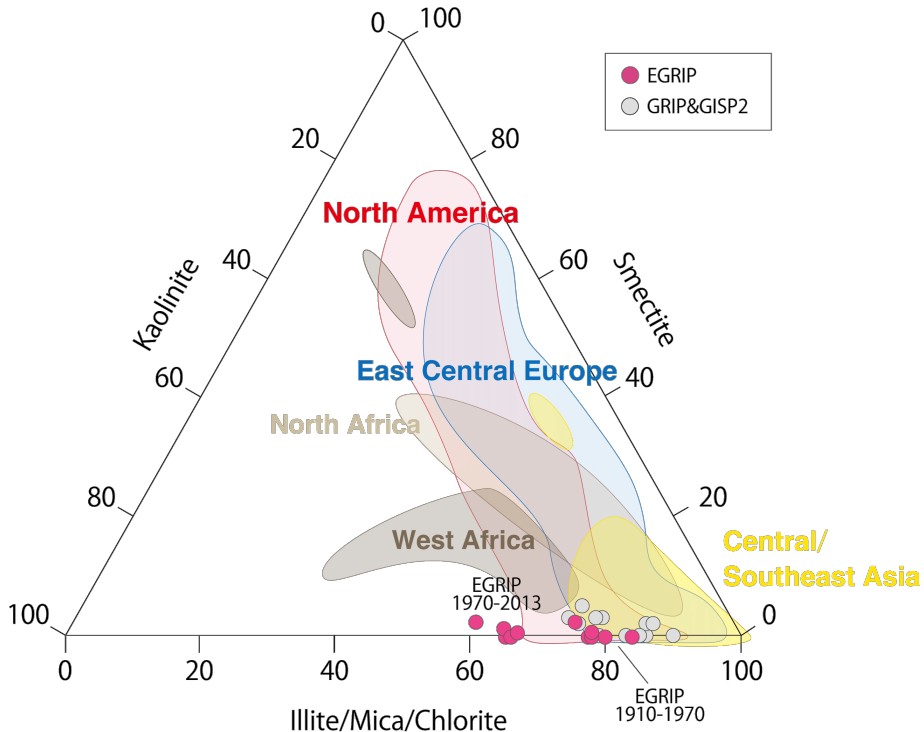

**Figure 9.** Ternary clay mineralogy diagram of EGRIP ice-core dust during period from 1910 to 2013 and published data for SIGMA-D ice-core dust (Nagatsuka et al., 2021) and Greenland Ice Sheet Project 2 (GISP2) and Greenland Ice Core Project (GRIP) ice-core dust during last glacial maximum and their potential source areas in illite/micas/chlorite-smectite-kaolinite space (GRIP and GISP2: Biscaye et al., 1997; Svensson et al., 2000, East Central Europe: Biscaye et al., 1997; Svensson et al., 2000; Újvári et al., 2012, 2015, 2022; Martinez-Lamas et al., 2020, North America: Potter et al., 1975; Biscaye et al., 1997; Svensson et al., 2000; Donarummo et al., 2003; Sionneau et al., 2008; Újvári et al., 2015, 2022, Central and Southeast Asia including Taklamakan Desert: Biscaye et al., 1997; Svensson et al., 2000; Újvári et al., 2015, 2022, Li et al., 2018, North Africa: Elmouden et al.,2005; Skonieczny et al., 2011; Újvári et al., 2022, West Africa including Sahara Desert: Chester et al., 1972; Svensson et al., 2000; Rodriguez-Navarr et al., 2018; Skonieczny et al., 2011; Újvári et al., 2022).



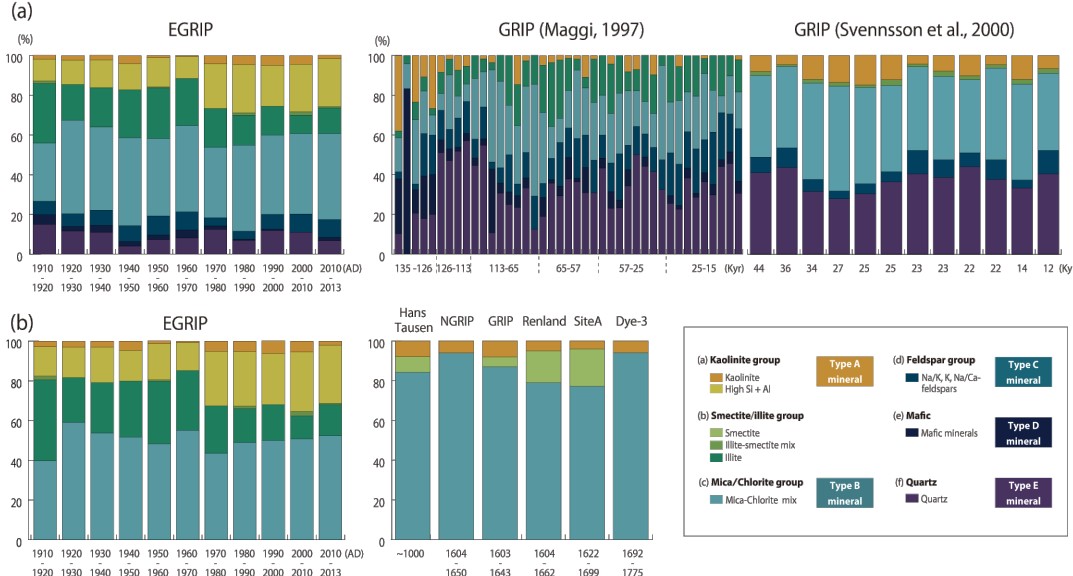

**Figure 10.** Comparison of (a) silicate mineral compositions between EGRIP and GRIP ice cores (Maggi, 1997; Svensson, 2000) and (b) clay mineral compositions between EGRIP and Hans Tausen, NGRIP, GRIP, Renland, Site A, and Dye-3 ice cores (Bory et al., 2003b). Note that Svensson et al. (2000) and Bory et al. (2003b) did not separate illite from mica/chlorite mix.

Compared to the silicate mineral compositions (GRIP, Svensson et al., 2000; GISP2, Maggi, 1997) and the clay mineral compositions (Hans Tausen, NGRIP, Renland, Site A, and Dye-3 sites, Bory et al., 2003b) of Greenland ice-core dust, which primarily originated from Asian deserts during glacial periods and the Holocene, the EGRIP ice core shows a relatively similar clay mineralogy (Fig. 10(b)). However, the EGRIP ice-core dust contains significantly lower amounts of Type E (quartz) mineral and higher amounts of illite, high-Si+Al and Type B (micas and chlorite) minerals (Fig. 10(a)). Direct comparisons of the EGRIP mineral composition records with previous studies are limited by the lack of continuous dust composition from other Greenland sites over the past 100 years. Additionally, differences in analysed periods (modern vs. mostly glacial), analytical techniques (X-ray diffraction vs. SEM-EDS), and mineral identification methods further complicate comparisons. X-ray diffraction analysis, used in previous studies, struggles to distinguish between micas and illite, whereas the elemental peak intensity ratio sorting scheme used in this study cannot separate micas from chlorite, leading to their identification as a mixed layer of micas/chlorite. Moreover, differentiating halloysite (a high-Si+Al mineral) from kaolinite requires specific chemical and physical treatments prior to X-ray diffraction analysis (e.g., Du Plessis et al., 2021). Consequently, previous studies may have identified illite as mica and high-Si+Al as kaolinite based on their X-ray diffraction analysis. These methodological differences likely explain the apparent absence of illite and high-Si+Al particles in the other Greenland ice-core dust samples. Despite these considerations, the significantly lower quartz abundance may indicate that the EGRIP ice-core dust likely originated not only from Asia, but also other sources. The quartz proportion in Saharan desert sand is lower than that in Asian desert sand (e.g., Svensson et al., 2000; Formenti et al., 2011), suggesting a greater contribution from African deserts to the EGRIP ice core compared to other Greenland ice cores from Last Glacial Maximum periods.



Previous studies have suggested additional source regions for Greenland ice-core dust. Regional climate modelling by Újvári et al. (2022) indicated a plausible European dust contribution to Greenland. EGRIP ice-core dust samples from 1910 to 1970 exhibit values in a ternary clay mineralogy diagram similar to those for east-central Europe. A trajectory analysis also indicates potential air mass transport from northern Eurasia to the EGRIP site (Fig. 11b). Thus, Europe is a possible source of EGRIP ice-core dust, although the absence of SEM-CL analyses for European dust prevents us from providing definitive

evidence of its transport. Modelled dust transport pathways to Greenland (Andersen et al., 1998; Mahowald et al., 2011) suggest that North America is also a potential source, and that North American glacial aerosols may have mixed with Asian dust during transit. On the other hand, Újvári et al. (2022) concluded that North America contributed minimally to the last glacial ice-core dust in central Greenland (GRIP and GISP2) based on isotope ratios and modelling results, despite some samples plotting close to North American loess in a ternary clay mineralogy diagram (Fig. 9). In the same diagram, EGRIP

ice-core dust samples from 1910 to 1970 exhibit similarities to the GRIP and GISP2 ice-core dust analysed by Újvári et al. (2022). Trajectory analyses also suggest potential air mass transport from North America to the EGRIP site (Fig. 11). Although SEM-CL analysis distinguishes EGRIP dust from Alaskan loess, we analysed only one sample from North America for SEM-CL, limiting our ability to draw definitive conclusions. Therefore, North America cannot be ruled out as an additional source.

**4.2 Changing contribution of ice-core dust from potential sources**

Although the mineral composition of the EGRIP ice-core dust has remained relatively stable over the past century, variations in specific mineral types suggest changes in dust sources during this period. The relative abundance of silicate minerals in the ice core shows differences among mineral types (Fig. 8).  Since the 1970s, the relative abundance of illite has decreased, while that of high-Si+Al minerals (Type A) has increased, indicating contributions from multiple geological sources with

varying intensities. In a ternary clay mineralogy diagram, the EGRIP ice-core dust samples from 1910 to 1970 are located in the bottom-right region, close to samples from other Greenland ice cores (GRIP and GISP2), as well as loess and desert sand from Asia and North America (Fig. 9). In contrast, samples from 1970 to 2013 are located in the bottom-middle region, reflecting lower illite/micas/chlorite content compared to that in earlier periods. Temporal variations are also observed in the cluster compositions of quartz particles in the ice-core dust. The sample from 1940 to 1950 is plotted close to samples from

the Gobi Desert in East Asia, while those since the 1970s (except for 1980 to 1990) are plotted closer to samples from the Sahara Desert in West Africa (Fig. 7). These findings suggest a shift in the contribution from primary dust sources since the 1970s, with a decrease in contributions from Asia and an increase from Africa in the last 50 years.

The elemental concentration ratios support this shift. The concentration of nssCa, along with elements such as Al and Fe in ice core samples, commonly serves as a proxy for terrestrial dust (e.g., Whitlow et al., 1992; Drab et al., 2002; Oyabu et al.,

2016; Sinnl et al., 2022; Miyamoto et al., 2022). In the EGRIP ice core, nssCa accounts for 85–100% of the total Ca concentration, indicating its primarily continental origin. nssK and nssMg also potentially originate from terrestrial dust (e.g., Dibb et al., 2007; Kuramoto et al., 2011; Nakazawa et al., 2021). Scatter plots show that the slopes of nssMg/Al, nssK/Al, nssCa/Al, and especially Fe/Al concentration ratios versus Si/Al are steeper for samples from 1980 to 2013 compared to earlier samples (Fig. 12). These findings support that the contribution rate from the primary potential sources has changed

since 1970−1980. Formenti et al. (2011) compared elemental ratios of mineral dust from Asian and African deserts. They found that the Sahara Desert (denoted PSA NAF-2 in Formenti et al., 2011) has higher Fe/Al ratio than that of the Gobi Desert (denoted PSA EAS-4 in Formenti et al., 2011). Therefore, the gradual increase in Fe/Al in the EGRIP ice core since the 1980s (Figs. 12 and 13) may reflect an increase in African dust contribution.





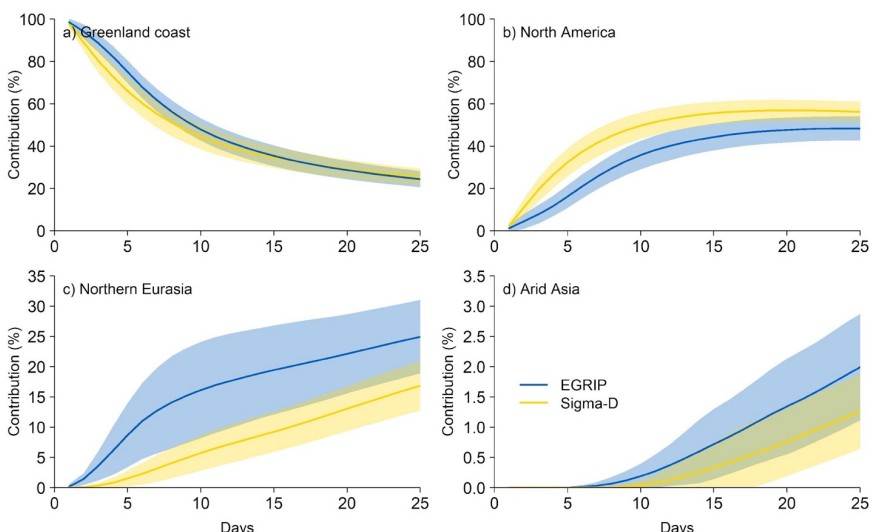

**Figure 11.** Air mass contribution (25-day back-trajectory) from (a) Greenland coast (GrC), (b) Northern Eurasia (EU and Russia), (c) North America (Canada and US), and (d) Arid Asia (Central Asia, Southeast Asia, and Middle East) to EGRIP and SIGMA-D sites.

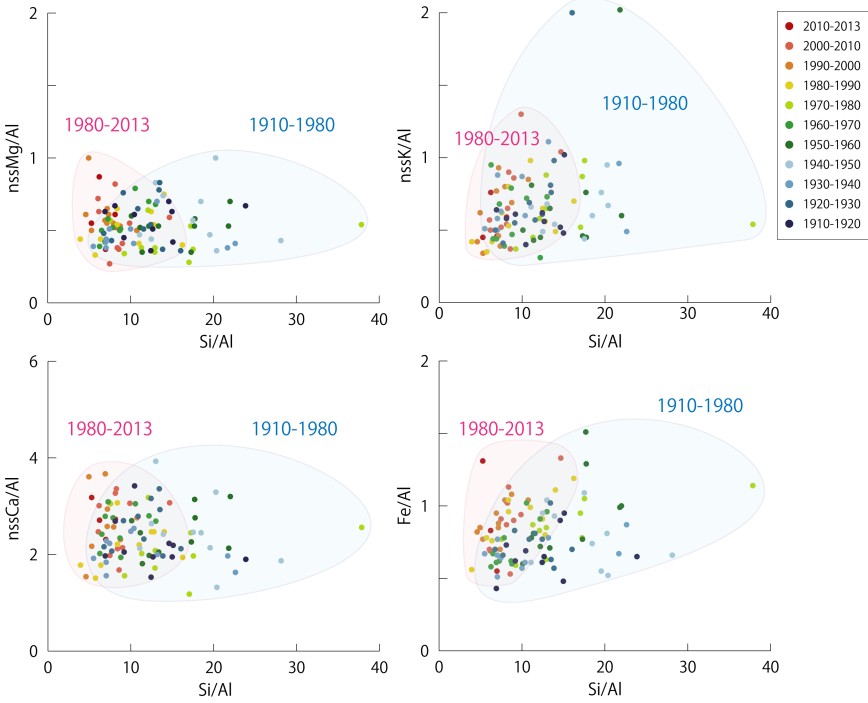

**Figure 12.** Annual average elemental concentration ratios of (a) nssMg/Al, (b) nssK/Al, (c) nssCa/Al, and (d) Fe/Al versus that of Si/Al in EGRIP ice core for samples from 1910 to 2013.





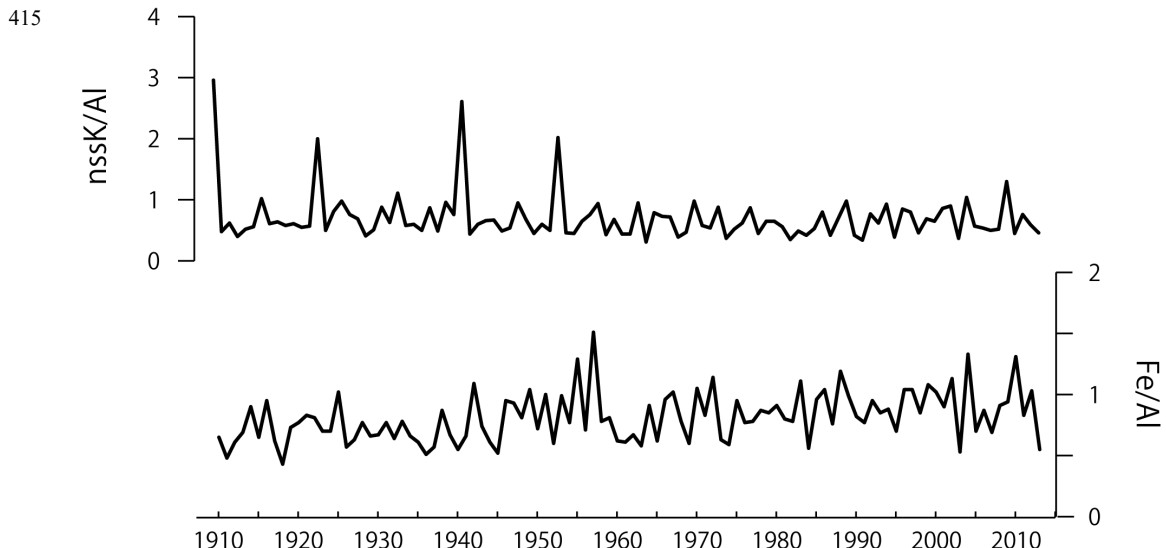

**Figure 13.** Historical records of annual average elemental concentration ratios (nssK/Al and Fe/Al) and annual average volume of dust particles in EGRIP ice core analysed by ICP-MS.

|  | NAO index | AO index | AMO index | PDO index |
|---|---|---|---|---|
| Total (1910-2013) | -0.23 | -0.16 | -0.39 | 0.17 |
| 1910-1950 | -0.12 | NA | 0.27 | 0.59 |
| 1950-1980 | -0.13 | 0.51 | -0.70 | 0.21 |
| 1980-2013 | -0.07 | -0.25 | -0.49 | 0.32 |

*NA: not available*

**Table 4.** 5-year running average correlation coefficient between average volume of EGRIP ice-core dust particles and NAO, Arctic Oscillation (AO), AMO, and Pacific Decadal Oscillation (PDO) indices





The shift in primary ice-core dust sources since the 1970s–1980s can be attributed to three possible factors: (1) variations in atmospheric circulations, (2) a decline in Asian dust events, and (3) increased local dust contribution.

(1) Atmospheric circulations: Previous studies have demonstrated the North Atlantic Oscillation (NAO) and the Atlantic Multidecadal Oscillation (AMO) likely affects atmospheric circulation patterns in the North Atlantic and dust transport to Greenland, including from Africa. For instance, during the negative phase of the AMO, characterized by a southward shift of the Intertropical Convergence Zone (ITCZ) and a cooling of the North Atlantic Ocean (e.g., Wang et al., 2012), or during the positive phase of the NAO, which features a stronger Azores high and a weaker Icelandic low (e.g., Clifford et al., 2019), surface winds intensify and precipitation in the southern Sahara decreases. These conditions lead to Sahelian droughts, enhancing Saharan dust transport to the North Atlantic and Mediterranean regions (e.g., Doherty et al., 2012, 2014). Prospero and Lamb (2003) also identified prolonged drought in the Sahara since the 1970s, driven by reduced precipitation, as a major factor contributing to increased Saharan dust transport across the Atlantic Ocean. These drought periods coincide with negative AMO and positive NAO phases (Fig. 14 (b) and (d)). Our mineralogical findings indicate that Saharan dust contributions to the EGRIP ice core have increased during these periods. Thus, these two atmospheric oscillations likely play a significant role in shifting primary dust sources of the EGRIP ice core.

Compared with atmospheric oscillation indices, the average volume of particles of the EGRIP ice-core dust (Fig. 14 (a)) calculated as the volume concentration divided by the number concentration, representing the distance from the source areas, exhibited a statistically significant negative correlation with the AMO index after 1950 (Table 4). The observed increase in ice-core dust volume during the negative phase of the AMO since the 1970s can be attributed to dust transport from Africa, which is geographically closer to Greenland than Asia. The back-trajectory analysis by Újvári et al. (2022) suggests that African dust travelled northward, mixed with European dust, and was transported westward into the Greenland interior during the Last Glacial Maximum. A similar route may have facilitated the transport of dust to EGRIP.

(2) Declining Asian dust events: Historical records indicate a rapid decline in dust events in East Asian deserts, particularly northern China, starting in the late 1970s and 1980s. The frequency of dust storms in East Asia after the 1980s has decreased to about half of the levels observed during the 1950s–1970s (Qian et al., 2002). This decline is primarily attributed to changes in meteorological conditions, especially a weakening of surface wind speed (e.g., Guan et al., 2017: Wu et al., 2018; Liu et al., 2020). The weakening of surface wind speed in the mid-latitudes of the Northern Hemisphere is likely linked to reduced meridional temperature gradients, a result of the polar amplification due to global warming (Wu et al., 2022). A similar trend has been recorded for a Tibetan ice core, where dust concentration began to decrease in the 1960s due to weaker dust emission strength (Huang et al., 2024). This reduction is also evident in the illite content of the EGRIP ice-core dust, which likely reflects a decline in Asian dust contribution. Elemental concentration ratios of the EGRIP ice core also highlight this change. Before the 1970s, the nssK/Al ratios showed multiple peaks with high concentrations, but such peaks have been less frequent since then (Fig. 13). The nssK is primarily sourced from clay illite and potassic feldspar (Formenti et al., 2011). Given that the EGRIP ice core contains little K-feldspar (Table 3 and Figs. 6 and 8), the lower nssK/Al ratios observed after the 1970s can be attributed to a reduction in illite, likely indicating a diminished contribution from Asian dust.

(3) Local dust contribution: Recent warming has reduced snow cover on the east coast of Greenland, likely decreasing snow cover duration (Fig. 14 (f) and (g)) and increasing local dust emissions. Enhanced local dust transport post-1980s has been reported, particularly along the coastal Greenland ice sheet (Amino et al., 2021; Nagatsuka et al., 2021) and even within the inland ice sheet (Kjær et al., 2022), including EGRIP site. Analyses of metallic compositions and particle size distributions in snow pit samples have revealed seasonal variations in the mineral dust sources at the EGRIP site: from winter to spring, most dust originates from distant arid sources, while from summer to autumn, a larger proportion comes from Greenland coastal regions (Komuro et al., 2024). However, the average volume of dust particles in the EGRIP ice core (Fig. 14(a)) has decreased since the 1980s and shows a positive correlation with modelled snow cover fraction anomalies during summer (June-August) on the southeast coast of Greenland (Fig. 14(f)), suggesting that the influence of local dust is likely small.



Additionally, the size and composition of the ice-core mineral dust also support a limited contribution of local materials in the EGRIP ice core. Since the ice-core samples analysed in this study are averaged over 10 years, the contribution of locally sourced dust is likely minimal, as summer dust concentrations at the EGRIP site are low (Komuro et al., 2024).

Thus, we concluded that the shift in the primary dust source since the 1970s–80s can be attributed to variations in atmospheric circulations and/or declining Asian dust events.

**4.3 Comparison with northwestern Greenland ice-core dust (SIGMA-D)**

The EGRIP ice-core dust samples mainly consisted of silicate minerals with small amounts of Fe-dominant minerals (Fig. 5), as was the case for the northwestern Greenland ice core (SIGMA-D, Nagatsuka et al., 2021). We did not identify any carbonate minerals in the EGRIP ice core. The EGRIP ice core was melted and filtered; the carbonate minerals in the ice core were probably dissolved in the melted ice samples. A comparison of the proportion of ice-core minerals other than

carbonate minerals shows no significant differences in the silicate and Fe-dominant minerals between the two ice cores (EGRIP: 86–97% and 2–6%, SIGMA-D: 73–96% and 1–16%, respectively). The lack of Al-dominant minerals in the EGRIP ice core might be due to differences in the geological sources.

The variability of the silicate mineral compositions differed largely between the EGRIP and SIGMA-D ice cores. The SIGMA-D ice-core dust shows multidecadal variations that are strongly influenced by changes in Greenland temperature

during the past 100 years (Nagatsuka et al., 2021). These variations are caused by an increased contribution of minerals originating from local ice-free areas, which is reflected in the abundance of Type B, C, D, and E minerals (Figs. 6 and 8). This increase resulted from shortened snow/ice cover duration in the Greenland coastal region during the warm periods of 1920s–1950s and 2000–2013 (Nagatsuka et al., 2021). In contrast, the EGRIP ice core shows significantly lower temporal variations with lower proportions of Type C, D, and E minerals (4–9%, 0–4%, and 4–12%, respectively) compared to those

in the SIGMA-D ice core (2–21%, 0–9%, and 3–22%, respectively) during warm periods. It has been suggested that dust sources depend on the distance of ice core sites from the ice-sheet margin and/or the altitude of the site. The majority of the dust deposited at interior sites (NGRIP, GRIP, Site-A, and Dye-3) is associated with long-range transport from distant deserts, whereas at coastal sites (Hans Tausen and Renland), the primary sources are local ice-free areas (Bory et al., 2003b). Therefore, the subtle compositional variation of the EGRIP ice core likely reflects a smaller local dust contribution compared

to that at the SIGMA-D site since the EGRIP site is located in a more interior region.

The morphological properties of the EGRIP ice-core dust also reflect a smaller contribution of local dust. The particle size distribution for EGRIP ice-core dust reveals lower mean and maximum diameters (0.85–1.36 μm and 5.59–8.94 μm, respectively, Table 2) with little variation compared to those for the SIGMA-D ice core (1.02–2.53 μm and 4.94–26.51 μm, respectively, Nagatsuka et al., 2021). According to Simonsen et al. (2019), particles with diameters smaller than 2 μm and

larger than 8 μm in Greenland ice cores can serve as indicators of distant and local sources of mineral dust, respectively. The SIGMA-D ice core contains coarse particles (larger than 8 μm), especially during warm periods when the local dust supply increased (1–9 particles in each sample, for a total of 150 particles, from 1915 to 1949 and from 2000 to 2013, Nagatsuka et al., 2021). In contrast, the EGRIP ice core has few particles with a diameter larger than 8 μm (a total of only 14 particles across all samples); the majority of particles are smaller than 2 μm. These findings suggest that the EGRIP ice-core dust

predominantly comprises particles transported over long distances, with a limited contribution of minerals from local sources.



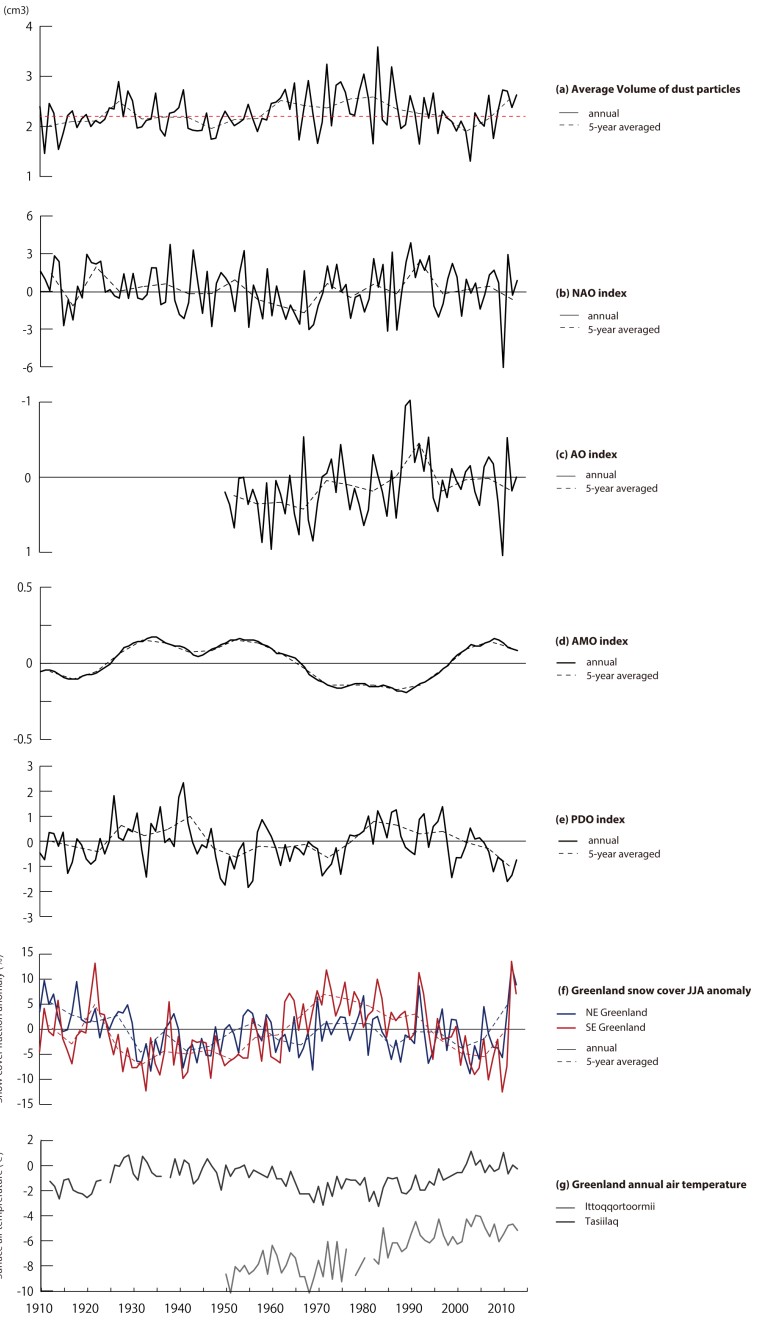

**Figure 14.** Historical changes in (a) annual average volume of EGRIP ice-core dust particles measured using Coulter counter, (b) NAO index (Hurrell and National Center for Atmospheric Research Staff, 2021), (c) Arctic Oscillation (AO) index (Climate Prediction Center website, http://www.cpc.ncep.noaa.gov/products/precip/CWlink/daily_ao_index/ao.shtml), (d) AMO index (Trenberth et al., 2023), (e) Pacific Decadal Oscillation (PDO) index (Japan Meteorological Agency website, https://ds.data.jma.go.jp/tcc/tcc/products/elnino/decadal/pdo.html), (f) snow cover fraction anomalies, and (g) surface temperature anomalies in Greenland. The temperature records for Ittoqqortoormii and Tasiilaq in eastern Greenland, located 750 km southeast and 1100 km south of the EGRIP site, respectively, are from Cappelen (2019). Snow cover fraction anomalies deviate from the 1915–2012 average in northeast and southeast Greenland. Horizontal dashed red lines in (a) indicate average values from 1910 to 2013.



The silicate mineral composition of the EGRIP ice core differs significantly from that of the SIGMA-D ice core (Figs. 6 and 8), indicating that the ice-core dust was likely transported from different geological sources. Nagatsuka et al. (2021) reported that the SIGMA-D ice core received substantial input of kaolinite (Type A), likely originating from relict deposits of warmer Tertiary climates in northern Canada during cold periods (1950–2004) over the past 100 years. Additionally, it was also supplied with Type C, D, and E minerals, mainly sourced from the Greenland coast during warm periods (1915–1949 and 2005–2013). In contrast, the EGRIP ice core showed a smaller proportion of kaolinite and Type C, D, and E minerals, but a higher proportion of Type B and illite content (1–4%, 10–21%, 21–37%, and 3–16%, respectively, Table 3 and Figs. 6 and 8) compared to those for the SIGMA-D ice core (5–66%, 9–34%, 3–25%, and 0–11%, respectively, Nagatsuka et al., 2021). The EGRIP ice core was likely primarily supplied with dust from Asian and African deserts, with additional contributions from Europe and North America, while local sources played a smaller role, as indicated by SEM-CL/EDS results (Section 4.1). Lower kaolinite and Type C, D, and E mineral content in the EGRIP ice core may reflect a smaller contribution from northern Canada and local sources, respectively. The higher illite content is possibly due to a larger contribution from Asian deserts to the EGRIP site than the SIGMA-D site. Újvári et al. (2015) analysed the clay mineralogy of potential source areas for Greenland ice cores and showed that Chinese loess has a higher illite proportion (64–71%) than Alaskan, central European, and Siberian loess (14–58%). There have also been reports of a large contribution of illite with a low contribution of smectite and a low kaolinite/chlorite ratio in Asian-sourced dust in Greenland ice cores and snow (e.g., Biscaye et al., 1997; Drab et al., 2002; Formenti et al., 2011; Újvári et al., 2022). These mineralogical characteristics are consistent with the EGRIP samples.

The back-trajectory results support the differences in the sources of ice-core dust between the EGRIP and SIGMA-D sites. To compare transport pathways of mineral dust from the EGRIP ice core with those from the SIGMA-D ice core, we applied the HYSPLIT back-trajectory model and calculated the probability distributions of an air mass from 1958 to 2014. Figure 2b shows the averaged probability distributions for an air mass arriving at the EGRIP site from 1958 to 2014, calculated from the 7-day back-trajectories. Figure 11 shows the regional contributions to the air mass arriving at the EGRIP (blue lines) and Sigma-D (yellow lines) sites in terms of backward temporal change. The HYSPLIT back-trajectory model suggests that the air mass mainly originated from Greenland (Figs. 2b, 11a). Excluding the ice sheet and ocean areas, which are not possible mineral dust sources, the air mass originated mainly from the Greenland coast (56–67%) as well as from North America (20–31%) and Northern Eurasia (6–20%, Figs. 11b and 11c). There is little inter-annual variation in the contributions from these three source regions. Between 1958 and 2014, most of the air mass at the EGRIP site originated from coastal Greenland, with a smaller proportion originating from North America and Northern Eurasia, including Europe and Russia (Fig. 11). These regions were also identified as potential dust sources for the SIGMA-D ice core (Nagatsuka et al., 2021). However, the relative contribution from North America and Northern Eurasia differed significantly between these two ice core sites. The contribution of air mass from North America to the EGRIP site was smaller, while that from Northern Eurasia was larger compared to the SIGMA-D site, which is consistent with the mineral composition results. Although mineralogical findings suggest that Asian and African deserts are likely primary sources, our trajectory results show little contribution from these regions, even in the 25-day back-trajectory. This is likely due to the inability of the back-trajectory analysis to identify dust transport from Asia to Greenland (Schüpbach et al., 2018). Therefore, accurately calculating the mineral dust contribution from potential source regions to the EGRIP site using our trajectory analysis is challenging.

## 5. Conclusions

We present the first continuous records of morphological and mineralogical properties of dust in the northeastern Greenland ice core (EGRIP) over the past 100 years, which were obtained using a multi-proxy provenance study focused on individual particle analysis. SEM-CL analysis on single quartz particles in the EGRIP ice core indicates that the primary sources of dust are likely Asian (Gobi) and African deserts (Sahara). Europe is likely an additional source and North America cannot be





ruled out as a potential source. These source regions are consistent with other interior Greenland ice-core dust from glacial periods. Furthermore, a ternary clay mineralogy diagram obtained from SEM-EDS analysis as well as the cluster composition of single quartz particles and the elemental concentration ratios indicate a shift in the contribution from primary dust sources since the 1970s−1980s, with a decrease in contributions from Asia and an increase from Africa in the last 50 years. This shift is likely linked to variations in atmospheric circulation (NAO and AMO) and the recent decrease in Asian

dust events.

The variability and mineral composition of the EGRIP ice-core dust differ significantly from those of the northwestern Greenland ice core (SIGMA-D). The SIGMA-D ice core shows multidecadal variations, with increased dust from the Greenland coastal region during warmer periods. In contrast, the EGRIP ice core shows much lower compositional variability, suggesting that its geological sources have remained relatively stable over the past 100 years. Given that most

particles are smaller than 2 μm and finer than those in the SIGMA-D core, the EGRIP dust primarily consists of particles transported over long distances, with only a minor contribution from local sources. Spatial variations in silicate mineral composition between the two ice cores suggests that the ice-core dust likely originated from different geological sources. The results suggest a smaller contribution from northern Canada and local sources, and a higher contribution from Asian dust, at the EGRIP site compared to the SIGMA-D site.

Although further analyses are needed to evaluate the contributions from North America and Europe to the EGRIP site, our findings demonstrate that SEM-CL analysis is a valuable tool for identifying ice-core dust sources and reconstructing their variations in ice cores with low dust concentration. This approach provides new insights into Greenland ice-core dust provenance and improves the prediction of variations in the darkening and ice mass loss of the Greenland Ice Sheet.



**Appendix A: Table**

**Table A1.** Six emission components (EC) determined for natural quartz particles and EC ratios for cluster 1-3 modified from Nagashima et al. (2023)

| | EC1 | EC2 | EC3 | EC4 | EC5 | EC6 |
|---|---|---|---|---|---|---|
| Center position (full width half maximum) (eV) | 1.75* | 1.9 (0.13)* | 2.0 (0.33)* | 2.2 (0.38)* | 2.7 (0.47)* | 2.9–3.2† |
| Related impurity/ imperfection | Substitutional $Fe^{3+}$ impurity centre* | Non-bridging oxygen hole centre* | Non-bridging oxygen hole centre* | Self-trapped exciton* | Self-trapped exciton or Ti impurity* | $Al^{3+}$ impurity centre* or imperfection centre§ |
| Cluster 1 | 22 ± 6% | 3 ± 1% | 38 ± 6% | 12 ± 2% | 10 ± 3% | 15 ± 6% |
| Cluster 2 | 22 ± 10% | 2 ± 1% | 19 ± 9% | 11 ± 3% | 16 ± 5% | 30 ± 8% |
| Cluster 3 | 4 ± 4% | 1 ± 1% | 8 ± 6% | 5 ± 2% | 21 ± 7% | 61 ± 11% |

\* Stevens-Kalceff et al., 2009 (with slight modification for centre position/full width half maximum).

† Not fixed because the precise nature of the centre has not yet been solved.

§ Götze et al., 2001 and references therein.



**Appendix A: Figures**

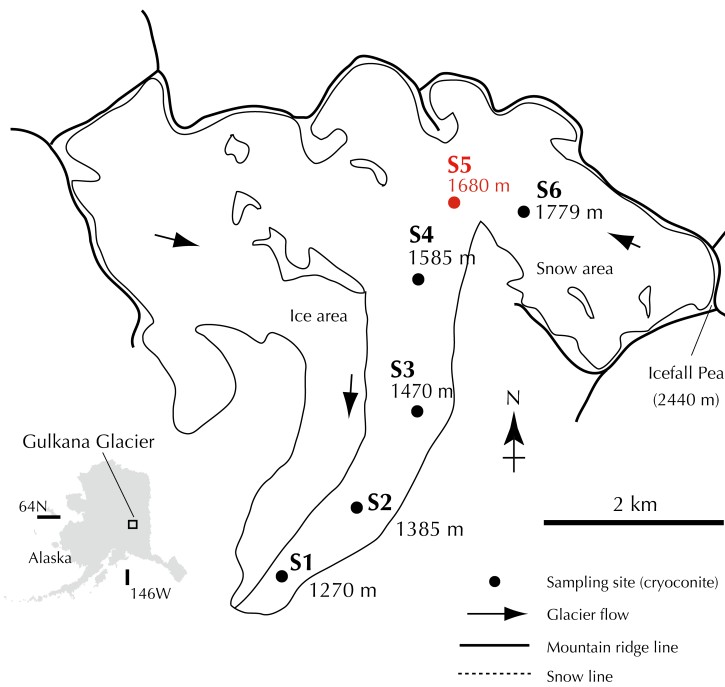

**Figure A1.** Location map modified from Nagatsuka et al. (2018), showing a sampling site of cryoconite for SEM-CL analysis from Gulkana Glacier in Alaska. We analysed quartz particles in a cryoconite sample from site S5, which had a bare ice surface and was located in the ablation area.



**Figure A2.** Na, SO₄, tritium, and ice layers (blue dashed line) records in upper 28.78 m (1880–2013) of EGRIP ice core (top). Major volcanic signals identified in SO₄ record are shown. Enlarged view from 22.6 to 23.725 m (bottom).






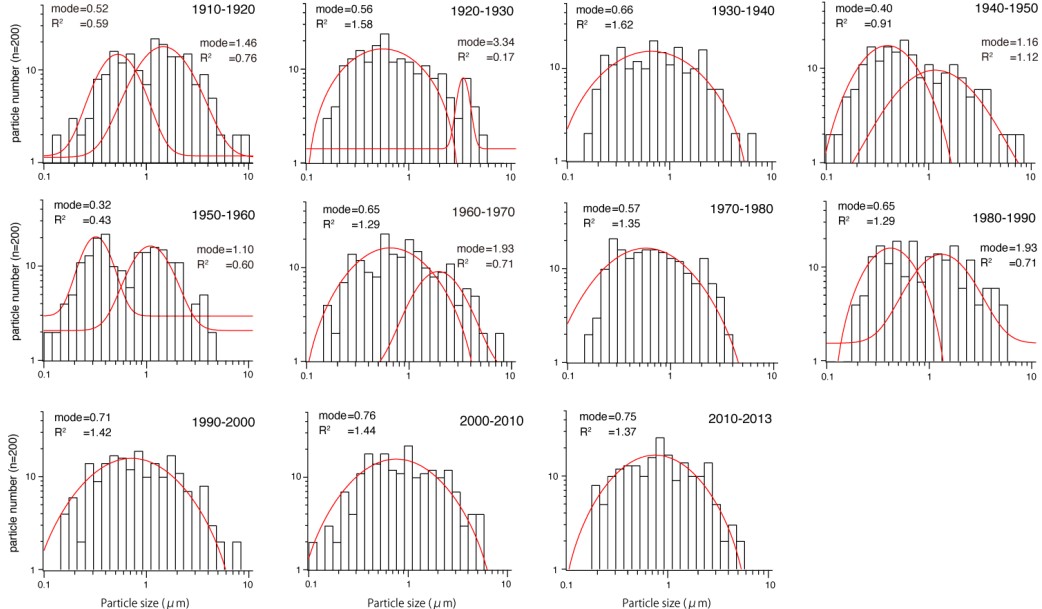

**Figure A3.** Comparison of size distribution and log-normal fitting results (mode: mode diameter; $R^2$: half peak width) for EGRIP ice-core mineral dust among samples.



*Data availability*

CMIP6 model output prepared for land-hist experiment is available at the link in the reference list (Malyshev et al., 2018;
NASA, 2018; Boucher, 2019; Danabasoglu, 2019; Stacke et al., 2019; Voldoire, 2019a and 2019b; Zhang et al., 2019;
Onuma and Kim, 2020; Peano et al., 2020; Wiltshire, 2020; Wiltshire et al., 2020).

Data on $\delta^{18}O$ and concentrations of sodium, sulfate (Na and $SO_4$), and tritium will be submitted to the ADS (Arctic Data
archive System) database for public use in further analysis.

*Author contributions*

NN designed the study and carried out the ice-core dust analysis and wrote the manuscript with the help of KGA, KN, KoF,
and FN. TJP and DDJ drilled the ice core. KN conducted the SEM-CL analysis. YK, MH, JO, KaF, YOT, KK, AY, and
KGA conducted the CFA analysis and data processing. YK, FN, and KGA conducted the Coulter counter analysis. YK,
KGA, FN, SOR, and GS determined the chronology of the ice core and YK also wrote the related paragraphs. NK conducted
the tritium measurement and wrote the related paragraphs. KoF conducted the back-trajectory analysis. YO conducted the
CMIP6 model analysis. DDJ led the EGRIP project. All authors discussed and commented on the paper.

*Competing interests*

The authors declare that they have no conflict of interest.

*Acknowledgements*

We would like to thank the members of the EGRIP projects for their generous support. EGRIP is directed and organized by
the Centre for Ice and Climate at the Niels Bohr Institute and US NSF, Office of Polar Programs. It is supported by funding
agencies and institutions in Denmark (A. P. Møller Foundation, UCPH), US (US NSF, Office of Polar Programs), Germany
(AWI), Japan (NIPR and ArCS), Norway (BFS), Switzerland (SNF), France (IPEV, IGE), and China (CAS). We sincerely
appreciate the SEM-CL measurement support provided by Sayuri Kubo. We also would like to thank Hiromi Okumura at
NIPR for supporting the Coulter counter analysis. This research has been supported by the JSPS KAKENHI Grant Numbers
JP18H04140, JP19K20443, JP20K19962, JP22J40168, the Arctic Challenge for Sustainability (ArCS) Project (Program
Grant Number JPMXD130000000)), the Arctic Challenge for Sustainability II (ArCS II) Project (Program Grant Number
JPMXD1420318865), the Integrated Research Program for Advancing Climate Models (Program Grant Number
JPMXD0717935457), the Advanced Studies of Climate Change Projection (Program Grant Number JPMXD0722680395),
the Environment Research and Technology Development Funds (JPMEERF20172003, JPMEERF20202003, and
JPMEERF20232001) of the Environmental Restoration and Conservation Agency of Japan.

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
