# Peer review of "A 100-Year record of mineralogical variations in Northeastern Greenland ice-core dust: Insights from individual particle analysis"

_EGUsphere, 2025_

## Author Comment (AC1)

We are grateful to Reviewer #1 for the thorough review and valuable comments on our manuscript. Our responses and the planned changes for the revision are explained below. Our replies are in blue and reviewer comments are written in black.

**Comments by Reviewer #1**

**1.** This paper attempts to characterise dust in the EGRIP ice core using mainly mineralogical methods. The data are new, covering around a century of dust input, and were probably the result of much painstaking work. I am reasonably happy with the dating as described. However, unfortunately, at high level my comments are the same as on previous versions of the paper. Even with the addition of CL data, the paper is unable to identify sources with any confidence. The statements in the abstract about changes in the relative proportions of Gobi and Sahara dust are simply not justified by the data in the text. The result is that the paper contains interesting data that is worth showing, but no gains in source apportionment that would justify publication in a climate journal such as CP. Additionally the paper is very long (with 14 figures) considering the low amount of new insight. I can only reiterate that a much shorter paper in a data journal might be more appropriate.

[**Reply 1-1**] Thank you very much for your insightful comments. We appreciate your recognition of the new data and the efforts involved in this work.

Regarding source attribution, we acknowledge that our previous manuscript did not provide sufficient detail to fully support our provenance conclusions. In the revised manuscript, we will increase the number of samples subjected to CL-analysis, conduct additional statistical analyses, and present uncertainties in cluster composition for each dataset in the revised manuscript. To address concerns about the relative contributions from the Gobi and Sahara Deserts, we have conducted further statistical analyses of elemental concentration ratios in the ice core (see Table Reply 1, details in Reply 1-22). These results support our interpretation of a provenance shift from Asian to African dust around the 1960s. Furthermore, we identified a statistically significant correlation between the average volume of ice-core dust and the AMO index since 1950s, which is known to strongly influence African dust emissions (Table Reply 2; details in Reply 1-23). We believe that these additional data and analyses will provide statistically robust support for identifying dust sources and assessing temporal changes in dust provenance.

If the reliability of the CL data can be fully demonstrated, our study would represent one of the few continuous records of Greenland ice-core dust provenance and its changes during a period of relatively low dust concentrations in recent decades. These findings offer new insights into Greenland ice-core dust provenance studies and contribute to paleoenvironmental reconstruction. Furthermore, identifying not only the source regions of the ice-core dust but also its physical and chemical properties (i.e., composition and particle size), and understanding their variability is crucial - not only for improving predictions of surface darkening and ice mass loss of the Greenland Ice Sheet (as discussed in Line 50-63 of the manuscript) but also for advancing our understanding of ice-nucleating particles (INPs), which play a critical role in cloud formation. Mineral particles often act as nuclei for ice crystal formation, and their effectiveness as INPs varies depending on the mineral type (e.g., Atkinson et al., 2013). For these reasons, we believe that the findings of this study provide valuable information suitable for publication a climate change journal. We will add this explanation to the revised manuscript to clarify the importance and novelty of our findings.

Finally, we agree with your comment regarding the manuscript length. To make the paper more concise and focused, we will remove Table 1 and Figures 10 and 11 and relocate some tables and figures (e.g., Figures 3 and 5) to the appendix, and eliminate any redundant content, thereby streamlining the manuscript. The reasons for removing these table and figures are explained in Reply 1-2-1, 1-4, and 1-16.

We hope these revisions will address your concerns and clarify the significance of our work.

Atkinson, J., Murray, B., Woodhouse, M. et al. (2013). The importance of feldspar for ice nucleation by mineral dust in mixed-phase clouds. Nature **498**, 355–358. https://doi.org/10.1038/nature12278.

**2-1.** The paper hinges on three types of data. First is the mineralogical data, as shown first in Figure 6. We are invited to interpret these in the light of Table 1. However this table only illustrates why this cannot be done, with each mineral class having numerous sources, and not in any way differentiating even at the continent scale.

[**Reply 1-2-1**] [Same as Reply 2-5 (Hereafter, Reply 2-XX indicates the comment number in our responses to Reviewer #2.)] We appreciate your insightful comment. In our previous submission (https://egusphere.copernicus.org/preprints/2023/egusphere-2023-1666/), one of our initial objectives was to compare the mineral compositions with those of the SIGMA-D ice core, as presented in Nagatsuka et al. (2021). For this reason, we adopted a similar mineral classification (Types A-E). However, as you correctly pointed out, this approach does not clearly differentiate the dust sources in the present study. Moreover, since the mineral classification does not significantly contribute to the main discussion of the manuscript, we will remove both the classification scheme and Table 1 in the revised version. Instead, to support the CL-based interpretation of the EGRIP ice-core dust sources and their temporal shifts, we will focus on variations in elemental concentration ratios that are considered indicative of minerals characteristic of Asian and African desert sources (details are provided in Reply 1-22).

**2-2.** The differences between samples from each decade at EGRIP are rather minor, and far from suggesting a change in source, to first order they reveal a constancy in mineral types (or at least the statistical analysis to establish otherwise is missing). For sure there are differences between EGRIP and SIGMA-D, but these are attributed mainly to the presence of more local material at the latter, an issue already discussed in a previous paper.

[**Reply 1-2-2**] Thank you for your valuable comments. We agree that the EGRIP ice core exhibits relatively minor differences in mineral composition among the samples. However, our additional statistical analyses indicate that the mineral composition at EGRIP varies significantly over time. Specifically, we performed a Chi-squared test on 11 age-defined samples (1910–2013) and 13 mineral types (Table 3). The results revealed statistically significant differences in mineral composition across different age groups ($p = 5.50 \times 10^{-10}$). Furthermore, to examine temporal variability in more detail, we divided the samples into two groups—pre- and post-1970, a period suggested as a potential shift in dust provenance by SEM-CL/EDS results—and conducted a PERMANOVA analysis. In this analysis, mineral types with five or fewer particles across all samples (smectite, U-D4, K-feldspar, Na/K-feldspar, Si-Mg, and Si-Fe) were excluded. The PERMANOVA results showed a statistically significant difference in mineral composition between the two time periods ($p = 0.0012$). These results suggest a shift in mineral composition around 1970, which may reflect a change in dust source regions.

Additionally, although differences between EGRIP and SIGMA-D have been attributed in part to local dust inputs at SIGMA-D, our data indicate systematic differences in the proportions of clay minerals likely formed under different environmental conditions. EGRIP samples contain higher amounts of mica/chlorite and illite, minerals typically formed by physical weathering, while SIGMA-D samples show higher kaolinite content,

indicative of chemical weathering. Since these minerals likely originate from distant sources, their contrasting abundances support the idea of distinct dust provenance.

**3-1.** The second line of evidence is the CL data, shown in Fig 7 as ternary plots. I found the data treatment for this work surprising. Clusters derived from Asian dust are used as a basis, and then interpreted in terms of relative proportions of Asian and African desert dust. This seems inappropriate. It is never explained where the Sahara data come from, and I cannot see how clusters appropriate for Asia can be used.

[**Reply 1-3-1**] Thank you for your valuable comment. We recognize that our explanation of Nagashima et al. (2023)—from which the three clusters of CL spectra were derived—and the meaning of CL spectra may not have been sufficiently clear, potentially causing some confusion. We will revise the text to clarify these points and provide a better explanation of how they relate to our study.

First, we would like to explain that the CL spectra of quartz particles reflect the conditions during quartz crystallization—such as temperature and growth rate—and the metamorphic pressure-temperature conditions experienced after crystallization (e.g., Zinkernagel, 1978). As a result, the CL spectra of quartz typically exhibit characteristic patterns depending on the parent rock type—namely, volcanic, plutonic, or metamorphic rocks (with slight variations depending on the degree of metamorphism) (e.g., Zinkernagel, 1978; Götze et al., 2001; Götte and Richter, 2006: Götze et al., 2021).

In Nagashima et al. (2023), the authors found that fine-grained quartz particles originating from Asia, Siberia, and Alaska generally fall into three distinct spectral types. These types are well correlated with empirically established CL characteristics associated with volcanic, plutonic, and low-grade metamorphic (or possibly sedimentary) rocks, as reported in earlier studies (e.g., Zinkernagel, 1978). Based on these findings, the three clusters used in this study do not represent the specific CL characteristics of Asian quartz alone, but rather reflect the typical CL signatures of three major rock types commonly found across the Earth's surface. We would like to provide further details on the samples used for the cluster analysis in Nagashima et al. (2023). The analysis included surface seawater samples from the Okhotsk Sea (four samples) and the Bering Sea (three samples), as well as quartz particles collected from the Gobi Desert (four samples) and the Taklimakan Desert (three samples). For the sampling locations used in the cluster analysis, please refer to Figure 1 in Nagashima et al. (2023): https://www.nature.com/articles/s41598-023-41201-6. The quartz particles in the seawater samples from the Okhotsk and Bering Seas are primarily transported by major rivers flowing into these seas—such as the Amur and Yukon Rivers—which collect sediments from their extensive drainage basins across regions including Siberia and Alaska. Therefore, the cluster analysis was not limited to samples originating from Asia alone.

[Same as Reply 2-4] The Sahara data are derived from three surface marine sediment samples collected at different latitudes along the northwestern African margin: #1 (26.82°N, 15.12°W), #2 (23.21°N, 17.85°W), and #3 (19.36°N, 17.28°W). Given the prevailing trade winds that transport Saharan dust westward, these samples are considered to reliably represent the average composition of Saharan dust sources. Detailed information on the CL data of these three reference samples is presented in a separate manuscript by our co-author Dr. Nagashima, which is currently under review in *Progress in Earth and Planetary Science* (PEPS). At the time of our preprint submission, this related work had not yet been submitted, and we could only cite it as "Nagashima et al., in preparation." We apologize for the lack of clarity this may have caused. We will cite the manuscript as appropriate in the revised version, depending on its progress (e.g., Nagashima et al., submitted / Nagashima et al., 2025).

Zinkernagel, U. in Contributions to Sedimentology Vol. 8. (1978). Cathodoluminescence of quartz and its application to sandstone petrology (eds. Füchtbauer, H., Lisitzyn, A. P., Milliman, J. D., Seibold, E.) 69 p (Schweizerbart Science Publishers).

Götze, J., Plötze, M. and Habermann, D. (2001). Origin, spectral characteristics and practical applications of the cathodoluminescence (CL) of quartz - A review. Mineral Petrol. 71, 225–250. https://doi.org/10.1007/s007100170040.

Götte, T., and Richter, D.K. (2006). Cathodoluminescence characterization of quartz particles in mature arenites: Sedimentology, 53, 1347–1359, https://doi:10.1111/j.1365-3091.2006.00818.x.

Götze, J., Pan, Y., & Müller, A. (2021). Mineralogy and mineral chemistry of quartz: A review. *Mineralogical Magazine, 85*(5), 639–664. https://doi.org/10.1180/mgm.2021.72.

**3-2.** We simply have no information on other Asian and African sources (previous isotopic work, as I recall, does not suggest the Gobi as the most probably Asian source), and the differences between decades, with only 5 decades sampled, are not convincingly different (again a statistical treatment would be needed to establish that the raw CL data, before clustering, are different for the measured decades).

[**Reply 1-3-2**] Previous isotopic studies have shown contrasting views regarding the sources of Greenland ice core dust. While Bory et al. (2003a) and Újvári et al. (2022) identified the Taklamakan Desert as the primary source with minimal contribution from the Gobi Desert, Biscaye et al. (1997) and Svensson et al. (2000) considered the Gobi Desert to be one of the major sources. Therefore, the Gobi Desert could plausibly be one of the main sources of dust in the EGRIP core.

[Same as Reply 2-4] Since SEM-CL analysis is highly time-consuming, we analyzed only five samples before and after 1970 to investigate the potential provenance shift around that period. However, we agree with the reviewer that it is important to statistically demonstrate differences in CL data between time periods. To address this, we plan to increase the number of analyzed samples from before 1970, conduct additional statistical analyses, and present uncertainties in cluster composition for each dataset in the revised manuscript.

Bory, A. J.-M., Biscaye, P. E., and Grousset, F. E.: Two distinct seasonal Asian source regions for mineral dust deposited in Greenland (NorthGRIP), Geophys. Res. Lett., 30, 1167, https://doi.org/10.1029/2002GL016446, 2003a.

Újvári, G., Klötzli, U., Stevens, T., Svensson, A., Ludwig, P., Vennemann, T., Gier, S., Horschinegg, M., Palcsu, L., Hippler, D. Kovács, J., Di Biagio, C. and Formenti, P.: Greenland ice core record of last glacial dust sources and atmospheric circulation. J. Geophys. Res. Atmo.s, 127, e2022JD036597, https://doi.org/10.1029/2022JD036597, 2022.

Biscaye, P. E., Grousset, F. E., Revel, M., Van der Gaast, S., Zielinski, G. A., Vaars, A., and Kukla, G.: Asian provenance of glacial dust (stage 2) in the Greenland Ice Sheet Project 2 ice core, Summit, Greenland, J. Geophys. Res., 102, 26765–26781, https://doi.org/10.1029/97JC01249, 1997.

Svensson, A., Biscaye, P. E., and Grousset, F. E.: Characterization of late glacial continental dust in the greenland ice core project ice core, J. Geophys. Res., 105, 4637–4656, https://doi.org/10.1029/1999JD901093, 2000.

**4.** Finally the back trajectories in Fig 2 really show us nothing, except that air masses can come in from all directions to EGRIP.

[**Reply 1-4**] Thank you for your comment. We acknowledge that Figure 2 may appear to show that air masses

arrive at EGRIP from all directions. However, differences in the contribution of North American air masses between the EGRIP and SIGMA-D sites are evident not only from the air mass contribution rates from each source region shown in Figure 11, but also from the back trajectories in Figure 2.

We agree with the previous reviewer's comment that air mass passage over a region does not necessarily mean that the dust emitted from that region is transported to the ice core sites. However, it is also certain that dust cannot originate from areas that air masses do not pass through, despite uncertainties in back-trajectory analyses.

Taking these uncertainties into account, the revised manuscript uses back trajectories not as definitive evidence of dust origins, but as supporting circumstantial evidence alongside other analytical results. Therefore, we respectfully believe that stating Figure 2 "shows us nothing" may understate its value.

Furthermore, following reviewer #2's suggestion, we will standardize the back-trajectory analysis to seven days, as 20-days back trajectories may lack reliability. Since Figure 11 was not intended to discuss daily variations in source contributions, it will be removed. Instead, we will present cumulative source contribution rates over seven days as pie charts in Fig. 2 (also shown in this reply as Figure Reply 1, with details in Reply 2-7).

For these reasons, we would prefer to retain Figure 2 as it currently stands.

[Figure]

Figure Reply 1. Map showing (a) the locations of the EGRIP and SIGMA-D ice core sites in Greenland, along with nine regions used to calculate regional contributions (GrIS: Greenland Ice Sheet, grey; GrC: Greenland coast, purple; NA: North America, orange; EU: Europe, blue; RUS: Russia, light blue; CA: Central Asia, yellow; SA: Southeast Asia, green; ME: Middle East, red; and AF: Africa, brown), and probability distribution for air mass at (b) EGRIP and (c) SIGMA-D sites from 7-day three-dimensional back-trajectory analysis from 1958 to 2014. The probability distribution was calculated by summing the air mass trajectories at 50, 500, 1000, and 1500 m above ground level. The pie charts illustrate the cumulative source contribution over the seven-day back trajectories from land regions other than Greenland, calculated according to the regional definitions shown in panel (a). The total contribution from all non-Greenland land areas is normalized to 100%, and the relative contributions of each region are presented. (modified from Figure. 2)

**5.** Overall, I do not feel that the paper has constructed a story that warrants a long paper like this. I will go into some details in the following paragraphs, as well as giving some minor points and typos. I apologise that this comes across negatively – that the authors have tried to construct a long discussion where the data do not support it, and this leads to a lot of unsupported statements.

[**Reply 1-5**] Thank you for your comment. In the revised manuscript, we will reorganize the overall structure to make the paper more concise and coherent. Additionally, we will thoroughly re-examine the basis of each of our claims. In particular, we will strive to clearly demonstrate the reliability of the data and the statistical significance of our findings in order to build a more persuasive argument.

**6.** Abstract "The SEM-CL/EDS results reveal that the primary dust sources in the EGRIP ice core are Asian (Gobi Desert) and African (Sahara Desert) deserts". I'm afraid I don't feel that the CL data showed this convincingly, and certainly the data are insufficient to justify the next sentence about their relative proportions.

[**Reply 1-6**] Thank you for your comment. As mentioned in Reply 1-1, we are increasing the number of samples analyzed by SEM-CL and will present the associated uncertainties in cluster composition for each dataset. These efforts aim to enhance the statistical robustness of our results and allow a more reliable assessment of regional differences. We will include these improvements in the revised manuscript.

**7.** Lines 44/45: surely GRIP is central Greenland and NEEM is north. The text says the opposite.

[**Reply 1-7**] Thank you for pointing this mistake. We will revise the description of the NEEM ice core from 'central' to 'north' in the revised manuscript.

**8.** Line 50 "Greenland ice-core dust can also be used to estimate variations in the mass balance of the ice sheet". This is incorrectly written. I think you mean that dust can influence the mass balance, not that the dust can tell us about the mass balance.

[**Reply 1-8**] Thank you for pointing this out. We will correct these sentences "Greenland ice-core dust can also influence the mass balance of the ice sheet."

**9.** Line 107 "nearly 100 years (1910–2013)". Nearly implies less than 100, but this is more than 100.

[**Reply 1-9**] We thank the reviewer for this remark. We will correct the sentences "over the past 100 years".

**10.** Fig 1. Hans Trausen should be Hans Tausen

[**Reply 1-10**] Thank you for your remark. We will correct this accordingly in the revised manuscript.

**11.** Line 205 and around. I do not follow the logic of using Nagashima's clusters for Asia to differentiate between Asian and African sources. You need surely to use source materials to define an African cluster

before attempting this analysis.

[**Reply 1-11**] As explained in Reply 1-3-1, three clusters used in our study represent quartz particles derived from different rock types (volcanic, plutonic, and low-grade metamorphic or possibly sedimentary rocks) commonly found across the Earth's surface and do not represent the specific CL characteristics of Asian quartz alone. Therefore, we consider it appropriate to apply these clusters in the source identification of African dust. This interpretation is further supported by the strong similarity in CL spectra between quartz particles from the Sahara Desert and the established clusters.

To address your concern more clearly, we will include the CL data of individual quartz particles and demonstrate the similarity statistically in the revised manuscript, thereby providing more robust support for our source identification.

**12.** I suspect it would also be important to look at the raw data (strength of each emission component) to ensure you are right that different Greenland samples are statistically significantly different before even attempting a cluster analysis.

[**Reply 1-12**] Thank you for your appropriate comment. Following your comment, we'll also show the difference using the CL data in the revised manuscript.

**13.** Fig 4. What is R2 in these figures. Presumably not a correlation coefficient as these are always less than 1!

[**Reply 1-13**] Thank you for your comment. R2 in the figure represents the Full Width at Half Maximum (FWHM) of the peak (mode) of the particle size distribution. However, since it is not particularly necessary in the discussion, we will remove it.

**14.** Line 300. Are the differences in mineralogy between decades statistically significant. My feeling is not but you need to address this.

[**Reply 1-14**] Thank you for your valuable comments. In response to your suggestion, we conducted additional statistical analysis to evaluate whether the differences in mineral composition between decades at EGRIP are statistically significant. As explained in Reply 1-2-2, these differences are statistically significant ($p < 0.05$), suggesting meaningful variability over time. Specifically, a PERMANOVA test comparing samples from before and after 1970—a period hypothesized to reflect a shift in dust provenance—reveals a statistically significant change in mineral composition between these two periods (PERMANOVA, $p < 0.05$). These findings support the interpretation of a shift in dust source regions around 1970. We will add these results and interpretations to the revised manuscript (details provided in Reply 1-2-2).

**15.** Section 3.4 and Fig 7. To me these look like tiny changes in composition. I also wonder why we are only shown 5 samples.

[**Reply 1-15**] Thank you for your valuable comments. As described in Reply 1-3-2, we initially analyzed only five samples to investigate a potential shift in dust provenance around the 1960s-1970s, given that SEM-CL analysis is highly time-consuming. In response to your suggestion, we will increase the number of samples subjected to CL-analysis and include estimates of uncertainties in cluster composition to statistically evaluate the significance of differences in the CL data in the revised manuscript.

**16.** Fig 6 and 8 (left) and 10 (EGRIP panel) seem to have the same information. How do they differ and why repeat the information?

[**Reply 1-16**] Thank you for your comment. As you pointed out, Figures 6 and 8 show similar SEM-EDS compositional data from this study and the SIGMA-D core. However, we believe the two figures serve distinct purposes: Figure 6 highlights overall compositional differences, while Figure 8 more clearly illustrates variations by mineral type. For this reason, we would prefer to retain both figures. On the other hand, we will remove Figure 10 and related sentences, as Reviewer #2 suggested the difficulty in evaluating the comparison due to methodological differences (Comment 6).

**17.** Section 4.1 really doesn't provide enough information to be definitive or convincing about sources or change of sources. I'd need to see a lot more source signature information, and an analysis of statistics of differences between decades.

[**Reply 1-17**] In response to your comment, we have taken several steps to strengthen the evidence regarding the dust sources and their temporal changes. First, we have conducted additional statistical tests, revealing that the mineral composition at EGRIP differs significantly across different time periods (A Chi-squared test and PERMANOVA, $p < 0.05$; details are provided in Reply 1-2-2). In addition, in a ternary clay mineralogy diagram (Figure 9), we plotted data points for each sample by source region, allowing us to narrow down the candidate areas (see Figure Reply 2). As a result, we found that the dust composition closely matched that of loess and desert regions in Southeast Asia, including the Gobi and Taklamakan, and also showed similarities to loess source regions in parts of the eastern United States and eastern Europe (Figure Reply 2; details in Reply 1-18). Finally, we will increase the number of samples subjected to CL-analysis to improve data reliability and present associated uncertainties, thereby clarifying the statistical significance of regional differences.

**18.** Fig 9 –I am not really sure what I am meant to get from this figure, since most of the source signatures overlap close to the Greenland data.

[**Reply 1-18**] Thank you for your comments. The most evident conclusion from Figure 9 is the clear shift in dust composition observed around the 1970s, which likely reflects a change in dust source regions. However, we agree with the reviewer that, in the previous version of Figure 9, the EGRIP ice core dust overlapped with multiple potential source regions, making it difficult to distinguish individual contributions. To address this, we revised the figure to plot the data points for each sample by source region, which allowed us to better constrain the candidate source areas (see Figure Reply2).

As a result, we found that the dust composition from 1910 to 1970 closely matched loess and desert regions in Southeast Asia, including Gobi and Taklimakan. In addition, for North America, six samples showed

similarities with Alaska and Canada, while one sample resembled values from each of Tennessee, Ohio, and Michigan. However, based on the results of the CL analysis, the values for Alaska differed significantly from those of the EGRIP dust, and thus can be excluded as potential sources. On the other hand, Ohio and Tennessee are extensive loess source regions (e.g. Muhs et al., 2001), suggesting that these areas may have contributed additional material. For Europe, only a few samples from limited regions in Eastern Europe, such as Serbia and Ukraine, showed similar values. Although we did not perform CL analysis on these samples and thus cannot confirm this directly, these regions are also known as major loess source areas (e.g., Buggle et al., 2008; Jary and Ciszek, 2013). Moreover, since Greenland ice-core samples (GRIP and GISP2) plotting close to EGRIP (Figure 9) have been suggested to contain a mixture of African dust and European loess, there is a strong possibility that these regions represent additional dust sources. We will include the above content in the revised manuscript.

Muhs, D. R., Bettis, E. A., III, Been, J. and McGeehin, J. P. (2001). Impact of Climate and Parent Material on Chemical Weathering in Loess-derived Soils of the Mississippi River Valley. Soil Sci. Soc. Am. J., 65: 1761-1777. https://doi.org/10.2136/sssaj2001.1761.

Buggle, B., Glaser, B., Zöller, L., Hambach, U., Marković, S. B., Glaser, I., & Gerasimenko, N. (2008). Geochemical characterization and origin of Southeastern and Eastern European loesses (Serbia, Romania, Ukraine). Quat. Sci. Rev., 27(9–10), 1058–1075. https://doi.org/10.1016/j.quascirev.2008.01.018.

Jary, Z., and Ciszek, D. (2013). Late Pleistocene loess–palaeosol sequences in Poland and western Ukraine. Quaternary International, 296, 37–50. https://doi.org/10.1016/j.quaint.2012.07.009.

**19.** Fig 10a GRIP panel – spelling of Svensson. Fig 10 needs a better caption, 10b has no information.

[**Reply 1-19**] Thank you for your comment. As explained in Reply 1-16, we will remove the figure.

**20.** Line 388 (and Fig 11). "North America cannot be ruled out as an additional source". But with the discussion as written nowhere can be ruled out.

[**Reply 1-20**] Based on the results of the CL analysis, the high-latitude regions of North America (Alaska) can be excluded as possible sources. We will add this point to the main text in the revised manuscript.

**21.** Line 402 "with a decrease in contributions from Asia and an increase from Africa in the last 50 years". I don't feel you established this.

[**Reply 1-21**] As described in detail in Reply 1-22, we conducted additional statistical analyses of elemental concentration ratios in the ice core, which suggest a shift in dust sources—from a dominant Asian contribution before the 1960s to an increasing influence from African sources thereafter (Table Reply 1). Furthermore, as mentioned in Reply 1-1, we plan to conduct additional CL analyses to further assess regional differences and to more robustly establish the shift in contributions from Asian to African dust sources.

**22.** Line 412 "Therefore, the gradual increase in Fe/Al in the EGRIP". I don't see the increase you are

considering. It's certainly not obvious in Fig 13, and Fig 12 is really unclear – what are the coloured shapes meant to show?

[**Reply 1-22**] Thank you for your comment. Our intention with the colored shapes in the Figure 13 was to highlight that the regression slope for samples from 1980–2013 is steeper than that for samples from 1910–1980. However, as you correctly pointed out, this trend is not immediately apparent. To clarify the temporal trend more clearly, we conducted additional linear regression analyses on elemental ratios (nssMg/Al, nssK/Al, Fe/Al, and nssCa/Al), dividing the data by decade (Table Reply 1).

Regarding the relationship between Fe/Al and Si/Al, no significant correlation is observed before the 1950s ($R^2 = 0.0002 - 0.2657$; see Table Reply 1). However, from the 1960s to 2000s, the correlation becomes stronger, ($R^2 = 0.4465 - 0.6734$) and regression slopes also increase over time, suggesting a strengthening association between higher Si/Al and Fe/Al values. The results imply a growing influence of Fe-rich sources, which could include North Africa. Conversely, the nssK/Al ratio showed a generally strong positive correlation with Si/Al during 1910–1950, except in the 1930s ($R^2 = 0.2385 - 0.6539$; see Table Reply 1), indicating a consistent input from a common dust source. After 1960, however, this correlation weakened significantly ($R^2 = 0.0029 - 0.3334$), and the slope became more variable. As discussed in the manuscript, nssK in the EGRIP ice core may be derived from illite, a clay mineral typically associated with Asian dust sources (line 453-455). Therefore, the significant correlation between K/Al and Si/Al during 1910–1950 likely reflects contributions from illite associated with Asian dust sources.

Taken together, these findings suggest a shift in the dominant dust source region around the 1960s—from Asian to African origin—which is consistent with the results suggested by SEM-CL/EDS. We will add the above results and the related table to the revised manuscript.

We created Figure 12 with the same intention as Figure 13; however, we agree that the increasing trend is not clearly visible. By combining Fe/Al with Si/Al, which represents a crustal origin, we believe the changes in mineral supply from different geological sources can be more clearly illustrated. Therefore, we have decided to remove Figure 12 and focus the discussion solely on Figure 13.

[Figure]

Figure Reply 2. Ternary clay mineralogy diagram of dust from the EGRIP ice core from 1910 to 2013, along with previously published data from potential source areas in the illite/micas–chlorite-smectite–kaolinite space. This is a revised version of Figure 9, in which data points for each region are plotted individually rather than enclosed in circles, in order to better constrain the candidate source areas    (East Central Europe: Biscaye et al., 1997; Svensson et al., 2000; Újvári et al., 2012, 2015, 2022; Martinez-Lamas et al., 2020, North America: Potter et al., 1975; Biscaye et al., 1997; Svensson et al., 2000; Donarummo et al., 2003; Sionneau et al., 2008; Újvári et al., 2015, 2022, Southeast Asia: Biscaye et al., 1997; Svensson et al., 2000; Újvári et al., 2015, 2022, Li et al., 2018).

**23.** Fig 14 and the surrounding discussion was really unconvincing. It doesn't seem to be related to sources in any case. As an example "a statistically significant negative correlation with the AMO index after 1950": while I see the values in Table 4, in Fig 14 when I compare panels a and d, I see no correlation.

[**Reply 1-23**] Thank you for your valuable comment. To clarify our interpretation, we additionally conducted Spearman's rank correlation analysis between the AMO index and the average ice-core dust volume across three distinct periods (1910–1950, 1950–1980, and 1980–2013), as shown in Table Reply2.

Our results reveal a marked temporal shift in correlation. A weak but statistically significant positive correlation was observed during 1910–1950 ($\rho = 0.33$, $p = 0.04$). After 1950, however, the relationship reversed: a strong negative correlation emerged during 1950–1980 ($\rho = -0.80$, $p < 0.001$), followed by a moderate but still significant negative correlation in 1980–2013 ($\rho = -0.42$, $p = 0.02$). Since the negative phase of the AMO is one of the factors that increases dust transport from Africa (Line 450), the statistically significant negative correlation between dust particle size in the ice core and the AMO during the period from 1960 to 2000 (when the AMO was negative) indicates a connection with dust transport from Africa. Therefore, we believe that Figure 14, along with the supporting discussion, does provide relevant insight into the source-related variability in dust deposition. We acknowledge that these trends may not be immediately evident from a visual comparison of Figures 14a and 14d; hence, we will include the correlation analysis in the revised manuscript for clarity.

Additionally, no significant correlation was found between the NAO index and dust volume for any of the periods, and we will note this explicitly in the revised text.

**24.** On the same discussion please clarify what average volume is. If it's really "the volume concentration divided by the number concentration", then it should be the volume of a dust particle, which would be of order 1 um^3, but in the figure it's shown as cm^3. You need to explain what this is much better and also discuss why you think it is related to distance from the source, which is also not immediately obvious.

[**Reply 1-24**] As you pointed out, the "average volume" refers to the volume concentration divided by the number concentration, and the unit will be corrected to μm³ in the revised manuscript.

This parameter reflects the average size of dust particles and may serve as an indicator of the transport distance from the source regions. In general, larger particles are heavier and tend to settle more quickly, making them less likely to be transported over long distances. Therefore, a larger average volume suggests a greater contribution from nearby sources. This interpretation is consistent with the findings of Bory et al. (2003b), who reported that fine particles transported over long distances were deposited at almost all elevated interior sites on the Greenland ice sheet, whereas coarser particles of likely local origin were observed only at coastal sites. We will add the above explanation to the revised manuscript.

**25.** Fig 13 claims to include "annual average volume of dust particles" but it doesn't.

[**Reply 1-25**] Thank you for your remark. We will remove "annual average volume of dust particles" from the figure caption.

Table Reply 1.    Summary of linear regression results for the annual average elemental concentration ratios of (a) nssMg/Al, (b) nssK/Al, (c) nssCa/Al, and (d) Fe/Al versus Si/Al in the EGRIP ice core samples from 1910 to 2013. $R^2$ indicates the coefficient of determination for each relationship with Si/Al. Highlighted values indicate correlations with $R^2 \geq 0.5$ and statistical significance ($p < 0.05$).

| Decade | | nssMg/Al | nssK/Al | nssCa/Al | Fe/Al |
|---|---|---|---|---|---|
| 1910s | Slope | 0.0033 | **0.1274** | -0.0321 | 0.0077 |
| | R² | 0.0184 | **0.6539** | 0.0882 | 0.0535 |
| 1920s | Slope | -0.0016 | **0.1180** | 0.0355 | 0.0007 |
| | R² | 0.0007 | **0.5807** | 0.0440 | 0.0002 |
| 1930s | Slope | -0.0006 | 0.0087 | -0.0134 | 0.0090 |
| | R² | 0.0013 | 0.0593 | 0.0609 | 0.2657 |
| 1940s | Slope | -0.0070 | **0.0981** | -0.0560 | -0.0168 |
| | R² | 0.0297 | **0.6148** | 0.1545 | 0.1723 |
| 1950s | Slope | -0.0006 | 0.0388 | 0.0163 | 0.0132 |
| | R² | 0.0011 | 0.2385 | 0.0408 | 0.0787 |
| 1960s | Slope | **0.0427** | -0.0078 | 0.0493 | **0.0571** |
| | R² | **0.6572** | 0.0079 | 0.0876 | **0.6734** |
| 1970s | Slope | 0.0024 | 0.0013 | 0.0041 | **0.0147** |
| | R² | 0.0277 | 0.0029 | 0.0030 | **0.5419** |
| 1980s | Slope | 0.0115 | 0.0223 | 0.0453 | **0.0368** |
| | R² | 0.1196 | 0.2066 | 0.1565 | **0.6098** |
| 1990s | Slope | -0.0396 | 0.0661 | 0.0424 | 0.0449 |
| | R² | 0.2525 | 0.3334 | 0.0147 | 0.4465 |
| 2000s | Slope | -0.0035 | 0.0761 | 0.0531 | **0.0687** |
| | R² | 0.0024 | 0.3968 | 0.0448 | **0.5208** |
| 2010s | Slope | -0.0255 | 0.0208 | -0.1692 | -0.1032 |
| | R² | 0.0216 | 0.0297 | 0.3336 | 0.1493 |

Table Reply 2. Spearman's rank correlation coefficients (ρ), corresponding t-values, and p-values between the average volume of EGRIP ice-core dust particles and the AMO and NAO indices for three time periods (1910–1950, 1950–1980, and 1980–2013). Statistically significant correlations (p < 0.05) are highlighted.

|  | AMO | | | NAO | | |
|---|---|---|---|---|---|---|
|  | ρ | t-value | p-value | ρ | t-value | p-value |
| 1910-1950 | **0.33** | 2.09 | 0.04 | -0.11 | -0.67 | 0.51 |
| 1950-1980 | **-0.80** | -6.94 | 0.00 | -0.22 | -1.18 | 0.25 |
| 1980-2013 | **-0.42** | -2.52 | 0.02 | -0.14 | -0.79 | 0.43 |

---

## Author Comment (AC2)

We are grateful to Reviewer #2 for the thorough review and valuable comments on our manuscript. Our responses and the planned changes for the revision are explained below. Our replies are in blue and reviewer comments are written in black.

**Comments by Reviewer #2**

**1.** The paper by Nagatsuka and colleagues provides analysis of the size and composition of mineral dust over the last 100 years based on particles collected from the EGRIP ice core in northern eastern Greenland. The work aims to present the obtained size and compositional datasets and provides analysis to determine the sources of dust in the ice core and their temporal variation. The results are discussed in comparison to data previously published and relying on another ice core in northwestern Greenland (SIGMA-D). The main result highlighted in the paper is the lower temporal variation of dust composition in EGRIP, suggesting more local dust sources compared to SIGMA-D.

The data from this work are very interesting and deserve to be published. On the other hand, the discussion and conclusions are not supported by the analysis, and no robust evidence are provided for the source identification. Detailed reasons for this are provided in the general comments.

[**Reply 2-1**] Thank you very much for your insightful comments. We appreciate your recognition of the value of our dataset and its potential contribution to the understanding of dust provenance in Greenland. We acknowledge, however, that the discussion and conclusions in the previous version of the manuscript lacked sufficient support from the presented data, and that our interpretations regarding dust provenance were not adequately substantiated. To provide the robust evidence for the ice-core dust source identification, we will increase the number of analyzed samples and include the uncertainties in cluster composition to statistically evaluate the significance of differences in the CL data in the revised manuscript.

**2.** My judgment is that in the present form the paper would require major revisions to narrow the scope and/or better support conclusions. My suggestion would be to present the data (Section 3) and provide a concise and objective analysis of elements resuming the relatively long discussion now in Section 4. A "data paper" of a "measurement report" type could be more suitable for this.

[**Reply 2-2**] Thank you for your valuable comments. The issue regarding the lack of supporting evidence for our conclusion was also raised by Reviewer #1 (Comments 1-3). To address this concern, we conducted additional statistical analyses on 11 age-defined samples (1910–2013) and 13 mineral types. The results revealed statistically significant differences in mineral composition across different age groups (Chi-squared test, p < 0.05). Furthermore, a PERMANOVA test comparing samples from before and after 1970—a period hypothesized to reflect a shift in dust provenance—reveals a statistically significant change in mineral composition between these two periods (PERMANOVA, p < 0.05). These findings support the interpretation of a shift in dust source regions around 1970, which is consistent with the results suggested by SEM-CL/EDS.

Furthermore, we also conducted additional statistical analyses on elemental ratios (details provided in Reply 2-8) as well as on the relationship between dust particle size and climate oscillation indices (AMO and NAO). Regarding the latter results, we briefly note that there is a statistically significant correlation between the average volume of ice core dust and the AMO index since the 1950s, which is known to strongly influence African dust emissions. For full details, please kindly refer to our response (Reply 1-23) and Table Reply 2 to Reviewer #1 (hereafter, Reply 1-XX indicates the comment numbers in our responses to Reviewer #1). Both results further support our interpretation based on CL analysis that the primary sources of ice-core dust

were likely the Asian and African deserts, and that their relative contributions may have shifted around the 1960s - 1970s. Thus, these statistically significant results provide additional support for the conclusions derived from our CL data. In the revised manuscript, we will incorporate these findings and provide a concise discussion.

To avoid long discussion in Section 4, we will remove the comparison with results obtained using different methods (see Reply 2-6).

**3.** I also note that in several points of the paper, including the abstract and the conclusions, the authors mention that "the findings demonstrate that the SEM-CL analysis is a valuable tool for identifying ice-core dust sources and reconstructing their variation during periods of low dust concentration". This kind of sentence mostly belongs to methodological papers dedicated to test and evaluate techniques, instead of this kind of papers that should focus on scientific interpretation of results based on well-established analysis tools. This kind of statement also question me about the most appropriate journal and targeted scope of the work.

[**Reply 2-3**] Thank you for your comment. As you correctly pointed out, such statements are indeed more appropriate for methodological papers. We included the sentence "The SEM-CL analysis is a valuable tool for identifying ice-core dust sources and reconstructing their variation during periods of low dust concentration" to highlight the novelty of our analytical approach. However, the primary aim of this study is not to establish or validate a new analytical method, but rather to identify the sources of ice-core dust and interpret their variations. Therefore, we will delete these sentences in the revised manuscript to better align with the scientific focus and scope of the paper.

We would like to emphasize that, if the reliability of the CL data can be fully demonstrated, our study would represent one of the few continuous records of Greenland ice-core dust provenance during recent decades - a period characterized by relatively low dust concentrations. These findings offer new insights into Greenland ice-core dust provenance studies and contribute to paleoenvironmental reconstruction. Furthermore, identifying not only the source regions of the ice-core dust but also its physical and chemical properties (i.e., composition and particle size), and understanding their variability is essential. This knowledge is crucial for predicting surface darkening and ice mass loss of the Greenland Ice Sheet (as discussed in Line 50-63 of the manuscript), and for advancing our understanding of ice-nucleating particles (INPs), which play an important role in cloud formation. Since the efficiency of mineral particles as INPs depends on their type (e.g., Atkinson et al., 2013), detailed mineralogical data are particularly valuable.

For these reasons, we believe that our findings provide important contributions appropriate for publication in a climate change journal. We will add this explanation to the revised manuscript to better clarify the significance and novelty of our study.

Atkinson, J., Murray, B., Woodhouse, M. et al. (2013). The importance of feldspar for ice nucleation by mineral dust in mixed-phase clouds. Nature **498**, 355–358. https://doi.org/10.1038/nature12278.

General comments

**4.** Section 4.1 lines 333-340 and Fig. 7: the analysis referring to Fig. 7 is not very robust. Only 5 points are considered from each decade and compared to unpublished data. Apart from the fact that unpublished datasets

are considered (data which however should be made available to the scientific community prior to be used as interpretation of other data) the five points from the EGRIP core are at the edges between the West Africa and East Asia data, without – to my reading of the plot - an emerging clear pattern.

[**Reply 2-4**] [Same as Reply 1-3-1] Thank you for your valuable comments. Regarding the SEM-CL analysis results, we acknowledge that our previous manuscript did not provide sufficient detail to support our provenance conclusions. The unpublished Saharan dust data are derived from three surface marine sediment samples collected at different latitudes along the northwestern African margin: #1 (26.82°N, 15.12°W), #2 (23.21°N, 17.85°W), and #3 (19.36°N, 17.28°W). Given the prevailing trade winds that transport Saharan dust westward, these samples are considered to reliably represent the average composition of Saharan dust sources. Detailed information on the CL data of these three reference samples is presented in a separate manuscript by our co-author Dr. Nagashima, which is currently under review in *Progress in Earth and Planetary Science* (PEPS). At the time of our preprint submission, this related work had not yet been submitted, and we could only cite it as "Nagashima et al., in preparation." We apologize for the lack of clarity this may have caused. We will cite the manuscript as appropriate in the revised version, depending on its progress (e.g., Nagashima et al., submitted / Nagashima et al., 2025).

[Same as Reply 1-3-2] Regarding the limited number of data points, we initially analyzed only five samples to investigate a potential shift in dust provenance around the 1960s-1970s, given that SEM-CL analysis is highly time-consuming. However, in response to your suggestion, we will increase the number of samples subjected to CL-analysis and include estimates of uncertainties in cluster composition to statistically evaluate the significance of differences in the CL data in the revised manuscript.

**5.** Fig. 8 and more in general the interpretation of the clustering analysis (Sect. 4.1 and 4.2): the categorization into Types A-E described in Sect. 2.4 refers to mineral types and associate them to potential sources, but those are very broadly defined, which makes difficult to understand their usefulness for the data interpretation.

[**Reply 2-5**] [Same as Reply 1-2-1] Thank you for your valuable comments. In our previous submission (https://egusphere.copernicus.org/preprints/2023/egusphere-2023-1666/), one of our initial objectives was to compare the mineral compositions with those of the SIGMA-D ice core presented in Nagatsuka et al. (2021), and therefore we adopted a similar mineral classification (Types A-E). However, as you have pointed out, this broad classification does not allow clear differentiation of dust sources in the present study. Moreover, since this mineral classification does not significantly affect the main discussion of the manuscript, we will discontinue the classification and removed Table 1 in the revised version. Instead, to interpret the clustering analysis in Sections 4.1 and 4.2, we focus not on such categorization, but rather on changes in the chemical components that are thought to originate from minerals associated with Asian and African deserts (details in Reply 2-8).

**6.** Section 4.1 lines 356-367, Fig. 10: the comparison against other datasets and the similarities/differences discussed, ultimately in relation to dust origin, is difficult to evaluate due to the methodological differences mentioned, potentially affecting the reasoning and the implications of identified mineralogical patterns

[**Reply 2-6**] We fully agree with your point regarding the difficulty in evaluating the comparison due to methodological differences. In response, we will remove lines 356–367 and Fig. 10 from the revised manuscript.

**7.** Back trajectory analysis: I am unsure how to read and interpret the analysis of back trajectories. These are performed considering 20-days, a choice that is questionable. Indeed, this is very long, also compared to an aerosol lifetime of few days in the troposphere. Apart from the trajectory duration, I am not sure to understand how to use the information of Fig. 11, also based on the results of Fig. 2 showing that there is not a preferential direction of back trajectory air masses – which somehow can be expected as all data for more than 50 years are put together. Also attention as there is an inconsistency: in Sect. 2.6 the 20-days trajectories are mentioned, while in Fig. 11 is 25 days. Instead, Fig. 2 shows 7-days trajectories.

[**Reply 2-7**] Thank you very much for your valuable comment. We apologize for the confusion caused by inconsistencies between the figures and the methods. In our previous study (Nagatsuka et al., 2021), we conducted long-term back trajectory analyses, which initially led us to present similar results in this manuscript. However, as you correctly pointed out, 20-days back trajectories may lack reliability. In the revised manuscript, we will standardize the analysis to 7-days back trajectories.

Since the original Figure 11 was not intended to discuss daily variations in contribution rates from possible source areas, it will be removed. Instead, we will present pie charts illustrating the cumulative contribution rates from different source regions over a seven-day period in Figure 2 (we provide as Figure Reply 1 in this response letter). The contribution rates shown in these pie charts are based on the cumulative source attribution of trajectories originating from land areas other than Greenland, calculated according to the regional classifications defined in Figure 2a. The total contribution from all non-Greenland land areas is normalized to 100%, and the relative contributions of each region presented.

While the coastal regions of Greenland show the highest overall contributions, excluding these, Sigma-D exhibits a prominent contribution from North America (NA), whereas EGRIP is characterized by relatively larger contributions from Europe (EU) and Russia (RUS). Although too small to be clearly shown in the figure, the contribution from Central Asia—including China (yellow region)—was 0.03% for EGRIP and 0.0025% for Sigma-D, indicating that the contribution to EGRIP was approximately ten times greater.

**8.** Section 4.2 lines 40-414, Figures 12 and 13: to my reading of Fig. 12 I do not see a clear pattern in the data points, which instead seems quite dispersed. The authors should provide some statistical analysis to support their interpretation. I also do not identify an increase in Fe/Al in Fig. 13 as stated, and more statistical analysis should be provided to support this statement (line 413).

[**Reply 2-8**] [Same as Reply 1-22] Thank you for your valuable comment. In figure 13, we intend to highlight that the regression slope for samples from 1980–2013 is steeper than that for samples from 1910–1980. However, as you correctly pointed out, this trend is not immediately apparent. To clarify the temporal trend more clearly, we conducted additional linear regression analyses on elemental concentration ratios (nssMg/Al, nssK/Al, Fe/Al, and nssCa/Al), dividing the data by decade (Table Reply 1).

Regarding the relationship between Fe/Al and Si/Al, no significant correlation is observed before the 1950s ($R^2 = 0.0002 – 0.2657$; see Table Reply 1). However, from the 1960s to 2000s, the correlation becomes stronger, ($R^2 = 0.4465 – 0.6734$) and regression slopes also increase over time, suggesting a strengthening association between higher Si/Al and Fe/Al values. The results imply a growing influence of Fe-rich sources, which could include North Africa. Conversely, the nssK/Al ratio showed a generally strong positive correlation with Si/Al during 1910–1950, except in the 1930s ($R^2 = 0.2385 – 0.6539$; see Table Reply 1), indicating a consistent input from a common dust source. After 1960, however, this correlation weakened significantly ($R^2 = 0.0029 – 0.3334$), and the slope became more variable. As discussed in the manuscript,

nssK in the EGRIP ice core may be derived from illite, a clay mineral typically associated with Asian dust sources (line 453-455).

Taken together, these findings suggest a shift in the dominant dust source region around the 1960s—from Asian to African origin—which is consistent with the results suggested by SEM-CL/EDS. We will add the above results and the related table to the revised manuscript.

We created Figure 12 with the same intention as Figure 13; however, we agree your comments that the increasing trend is not clearly visible. By combining Fe/Al with Si/Al, which represents a crustal origin, we believe the changes in mineral supply from different geological sources can be more clearly illustrated. Therefore, we have decided to remove Figure 12 and focus the discussion solely on Figure 13.

[Figure]

Figure Reply 1. Map showing (a) the locations of the EGRIP and SIGMA-D ice core sites in Greenland, along with nine regions used to calculate regional contributions (GrIS: Greenland Ice Sheet, grey; GrC: Greenland coast, purple; NA: North America, orange; EU: Europe, blue; RUS: Russia, light blue; CA: Central Asia, yellow; SA: Southeast Asia, green; ME: Middle East, red; and AF: Africa, brown), and probability distribution for air mass at (b) EGRIP and (c) SIGMA-D sites from 7-day three-dimensional back-trajectory analysis from 1958 to 2014. The probability distribution was calculated by summing the air mass trajectories at 50, 500, 1000, and 1500 m above ground level. The pie charts illustrate the cumulative source contribution over the seven-day back trajectories from land regions other than Greenland, calculated according to the regional definitions shown in panel (a). The total contribution from all non-Greenland land areas is normalized to 100%, and the relative contributions of each region are presented (modified from Figure. 2).

Table Reply 1. Summary of linear regression results for the annual average elemental concentration ratios of (a) nssMg/Al, (b) nssK/Al, (c) nssCa/Al, and (d) Fe/Al versus Si/Al in the EGRIP ice core samples from 1910 to 2013. R² indicates the coefficient of determination for each relationship with Si/Al. Highlighted values indicate correlations with R² ≥ 0.5 and statistical significance (p < 0.05).

| Decade | | nssMg/Al | nssK/Al | nssCa/Al | Fe/Al |
|---|---|---|---|---|---|
| 1910s | Slope | 0.0033 | **0.1274** | -0.0321 | 0.0077 |
| | R² | 0.0184 | **0.6539** | 0.0882 | 0.0535 |
| 1920s | Slope | -0.0016 | **0.1180** | 0.0355 | 0.0007 |
| | R² | 0.0007 | **0.5807** | 0.0440 | 0.0002 |
| 1930s | Slope | -0.0006 | 0.0087 | -0.0134 | 0.0090 |
| | R² | 0.0013 | 0.0593 | 0.0609 | 0.2657 |
| 1940s | Slope | -0.0070 | **0.0981** | -0.0560 | -0.0168 |
| | R² | 0.0297 | **0.6148** | 0.1545 | 0.1723 |
| 1950s | Slope | -0.0006 | 0.0388 | 0.0163 | 0.0132 |
| | R² | 0.0011 | 0.2385 | 0.0408 | 0.0787 |
| 1960s | Slope | **0.0427** | -0.0078 | 0.0493 | **0.0571** |
| | R² | **0.6572** | 0.0079 | 0.0876 | **0.6734** |
| 1970s | Slope | 0.0024 | 0.0013 | 0.0041 | **0.0147** |
| | R² | 0.0277 | 0.0029 | 0.0030 | **0.5419** |
| 1980s | Slope | 0.0115 | 0.0223 | 0.0453 | **0.0368** |
| | R² | 0.1196 | 0.2066 | 0.1565 | **0.6098** |
| 1990s | Slope | -0.0396 | 0.0661 | 0.0424 | 0.0449 |
| | R² | 0.2525 | 0.3334 | 0.0147 | 0.4465 |
| 2000s | Slope | -0.0035 | 0.0761 | 0.0531 | **0.0687** |
| | R² | 0.0024 | 0.3968 | 0.0448 | **0.5208** |
| 2010s | Slope | -0.0255 | 0.0208 | -0.1692 | -0.1032 |
| | R² | 0.0216 | 0.0297 | 0.3336 | 0.1493 |

Detailed comments

**9.** Section 2.7, lines 223-226: this kind of sentences is not necessary / relevant

[**Reply 2-9**] Thank you for pointing this out. We agree with your comments and will delete the sentences.

**10.** Section 2.7, lines 230-235: this part can stay in supplementary information

[**Reply 2-10**] Thank you for your comment. We understand your suggestion to move the description of the CMIP6 models (lines 230–235) to the supplementary information. However, since Figure 14 - which uses snow cover fractions derived from these models—is presented in the main manuscript and discussed in the context of provenance shift of the ice-core dust, we believe a brief explanation of the models is necessary within the main manuscript. We recognize that your comment aims to reduce redundancy, so we will revise this section by removing unnecessary details to shorten the manuscript. Additionally, essential supporting information regarding the CMIP6 model data will be moved to the Data Availability section.

**11.** Section 2.7, lines 236-237: this is relevant to the analysis of the present paper and should be better detailed

[**Reply 2-11**] Thank you for pointing this out. We will add explanations how to calculate snow cover fraction anomalies in the northeast and southeast areas.

**12.** Section 3.2: the diameter is a projected-area diameter – please provide more details

[**Reply 2-12**] The projected-area diameter, referred to as the "equivalent circle diameter" in the manuscript, was measured using image-processing software (ImageJ, National Institutes of Health, USA) and represents the diameter of a circle with an area equivalent to the two-dimensional projection of the particle.

In contrast, particle size data shown in Figure 14 were obtained using a Coulter counter, which measures the equivalent spherical diameter in three dimensions. Therefore, we will also add a note explaining that the definition of particle size differs between the SEM-based measurements and those obtained using the Coulter counter.

**13.** Figure 4: y-axis should read "particle number concentration, x-axis should refer to "diameter" or "projected-area diameter" (if this is the case)

[**Reply 2-13**] Thank you for your comment. The Y-axis represents the number of particles (count number), not number concentration (We count 200 particles per sample, with 11 samples in total, making 2200 particles). We will revise the X-axis into "projected-area diameter", or "equivalent circle diameter" as described in the manuscript.